# Investigating the celerity of propagation for small perturbations and dispersive sediment aggradation under a supercritical flow

Hasan Eslami[1], Erfan Poursoleymanzadeh[1,2], Mojtaba Hiteh[1,3], Keivan Tavakoli[1,4], Melika Yavari Nia[1,5], Ehsan Zadehali[1,6], Reihaneh Zarrabi[1,7], Alessio Radice[1]

[1]Dept. of Civil and Environmental Engineering, Politecnico di Milano, Milan, 20133, Italy
[2]Dept. of Civil Engineering and Management, University of Twente, Enschede 7522NB, The Netherlands
[3]Dept. of Civil and Coastal Engineering, University of Florida, Gainesville, FL 32611, US
[4]Dept of Geography and the Environment, University of Texas at Austin, Austin, TX 78712 – 1697, US
[5]Dept. of Soil, Water and Ecosystem Sciences, University of Florida, Gainesville, FL 32611, US
[6]Dept. of Civil, Environmental, and Geo-Engineering, University of Minnesota Twin Cities, Minneapolis, MN 55455, US
[7]Dept. of Geography and Environment, University of Alabama, Tuscaloosa, AL 35487, US

*Correspondence to*: Alessio Radice (alessio.radice@polimi.it)

**Abstract.** The manuscript presents an investigation of the scales of propagation for sediment aggradation in an overloaded channel. The process has relevant implications for land protection, since bed aggradation reduces channel conveyance and thus increases inundation hazard; knowing the time needed for the aggradation to take place is important for undertaking suitable actions. Attention is here focused on supercritical flow, under which the process is dispersive and a depositional front cannot be clearly recognized; in these conditions, one needs to define propagation scales locally and instantaneously. Based on spatial and temporal rates of variation of the bed elevation, we quantify a celerity of propagation for the sediment aggradation wave. Furthermore, considering that morphological processes are modeled by a system of differential equations, the eigenvalues of the latter are the celerities of the so-called small perturbations. After a review of existing approaches to determine the celerity of small perturbations taking into account or discarding the concentration of transported sediment, the manuscript considers a laboratory experiment with temporally and spatially detailed measurements, whose results are representative of those of several others performed in the same campaign. The relationships between the local and instantaneous Froude number, celerity of small perturbations, and the celerity of the aggradation wave are explored. The celerity of the aggradation wave is correlated to that of the small perturbations, while their values differ by orders of magnitude. Our results indicate that accounting or not for the solid concentration in the governing equations does not significantly impact the correlation between the two types of celerity, even if one of the eigenvalues changes significantly in value. Finally, the aggradation celerity is generally below 0.05 times the initial flow velocity, this serving as a rule-of-thumb estimation that may be useful for engineering purposes.

## 1 Introduction

River sediment transport is one of the key processes shaping the Earth's surface and has a number of implications for human life (Dotterweich, 2008; Mazzorana et al., 2013; Haddadchi et al., 2014; Longoni et al., 2016; Pizarro et al., 2020). The morphologic evolution of rivers can be studied at a huge range of scales, the longest and shortest ones being related to geology/geomorphology and particle mechanics, respectively (e.g., Aksoy and Kavvas, 2005; Ancey, 2020, respectively). Within relatively short time scales, such as those for flash floods, bed aggradation may be induced by an imbalance between an amount of supplied sediment and the transport capacity of the flow. It is well established that, in such extreme events, sediment aggradation can cause a rapid rise in the riverbed elevation, thereby reducing the channel's conveyance. Therefore, in turn, the aggradation process may increase hydraulic hazard during a calamitous event (Sear et al., 1995; Stover and Montgomery, 2001; Lane et al., 2007). For example, Neuhold et al. (2009) studied the Ill river in Austria and reported that incorporating sediment transport into hazard assessment increases the probability of dyke overtopping; Radice et al. (2013) performed a back analysis of a past event for the Mallero river (Italy), which induced aggradation of up to 4 m in an in-town reach with 5-m banks. Pender et al. (2016) studied the Caldew river in England and showed how channel aggradation can

provoke flooding for events that would not induce it in the absence of morphological changes. Besides the magnitude of morphologic changes, the scales of the progressive evolution of the process are also important. In fact, when a channel is overloaded with sediment, an aggradation wave propagates with a certain celerity along the reach. Knowing the celerity of propagation is crucial for estimating when an aggradation wave will reach any key spot along the stream, thus increasing locally the hydraulic hazard. Again, this knowledge is particularly valuable for engineering purposes, as it enables better planning and risk assessment in managing sediment-related changes in river systems.

Sediment aggradation has been studied in the past for both the effects of pulsed sediment supply (e.g., Cui et al., 2003; Cui and Parker, 2005; Sklar et al., 2009) and the formation of depositional fronts (e.g., Soni, 1981; Yen et al., 1992; Alves and Cardoso, 1999; Zanchi and Radice, 2021). Cui et al. (2003) and Cui and Parker (2005) found that sediment entering into mountain stream in pulses causes topographic disturbances that are eliminated through translation (downstream movement) and dispersion (gradual fading). They discovered that pulse migration was primarily dispersive, although both translation and dispersion occurred when pulses supplied finer sediment than the surrounding material. Sklar et al. (2009) also found that sediment pulses on an armored bed evolved through both translation and dispersion. Furthermore, Soni (1981), Yen et al. (1992) and Alves and Cardoso (1999) provided quick predictors of a bulk celerity for a formed aggradation front; these formulae may be used to provide expeditious estimates of the time an aggradation wave would need to move from a sediment source to a critical point. Aggrading fronts induced by sediment overloading may be either translational or dispersive (Lisle et al., 2001). A translational front appears as a sharp discontinuity in the bed elevation (Fig. 1a). In a dispersive process, instead, a depositional wedge becomes progressively longer and thinner downstream (Fig. 1b). Zanchi and Radice (2021), based on laboratory experiments in subcritical conditions, could estimate the celerity of the aggradation wave by applying a tracing method since the sediment front was translational and thus reliably detectable in the performed experiments. They argued that translational features are favored by low Froude number and high overloading ratio (the ratio between the sediment inflow discharge and the initial sediment transport capacity of the flow), the opposite holding for dispersive features. In a supercritical flow, given a relatively high value of the Froude number, one expects a dispersive process to take place. In such a condition, due to the absence of a sharp sediment front, it is not possible to track the latter to estimate the celerity of propagation of the aggradation wave. Therefore, alternative methods are needed.

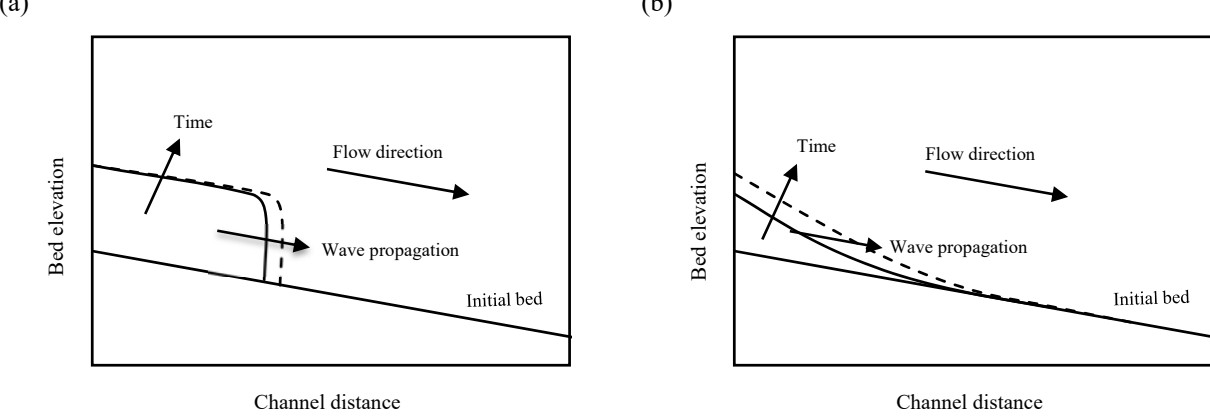

**Figure 1. Schematic representation of propagation of sediment aggradation wave for (a) translational front (b) dispersive process.**

From a mathematical point of view, several studies have been performed to simulate the sediment transport process in rivers by using a quasi-two-phase approach in a one-dimensional framework (e.g., De Vries, 1965, 1971, 1973, 1993; Armanini and Di Silvio, 1988; Sieben, 1997, 1999; Kassem and Chaudry, 1998; Lyn and Altinakar, 2002; Lee and Hsieh, 2003; Rosatti et al., 2005; Rosatti and Fraccarollo, 2006; Goutière et al., 2008; Armanini et al., 2009; Garegnani et al., 2011; Armanini, 2018). In a quasi-two-phase approach, the hydro-morphologic evolution of the bed and water surfaces is depicted by a hyperbolic system of partial differential equations, including mass and momentum conservation equations for the mixture phase and one continuity equation (the Exner equation) for the sediment phase. Furthermore, several authors have argued that the eigenvalues

of the hyperbolic system represent the celerity of propagation of "small perturbations" in bed and water. These "small" or "infinitesimal" disturbances are waves with limited amplitude generated when the surface of bed or water are perturbed locally (e.g., Rosatti et al., 2004). A classic example of such small-scale, local waves is that of the concentric ripples formed when a stone is thrown into water, propagating in different directions (e.g., Çengel and Cimbala, 2006). The volumetric concentration of the transported sediment, defined as the ratio of sediment discharge to the water-sediment mixture discharge, plays an important role in the formulation of the system of equations and in the determination of its eigenvalues. In fact, when this concentration is negligible, the system may be transformed into another one containing mass and momentum conservation equations for the clear water (i.e., the Saint-Venant equations) and the Exner equation. De Vries (1965) proposed to consider the sediment concentration negligible below a threshold value of around 0.002, while Garegnani et al. (2011, 2013) introduced a higher threshold of 0.01. Additionally, some studies have proposed approximate determinations of the eigenvalues of a simplified system of equations for negligible sediment concentration (De Vries, 1965, 1971, 1973; Lyn, 1987; Lyn and Altinakar, 2002; Goutière et al., 2008; Armanini, 2018). By contrast, Morris and Williams (1996) argued that the assumption of a negligible concentration is not appropriate for many natural streams and, based on that, provided an equation to estimate the eigenvalues of the full system of equations as the celerity of small perturbations.

An aggradation wave is a large-scale process that cannot be considered as a small perturbation, as a flood wave is much larger than the small ripples mentioned above. The celerity of propagation of this large-scale phenomenon may thus differ from that of small perturbations, because different processes can propagate at different velocities. As a result, one would reasonably expect that the aggradation wave celerity cannot be quantified by the eigenvalues of the system of governing equations. Determining the celerity of an aggradation wave necessitates a different quantitative approach.

Based on the arguments above, the present manuscript considers sediment aggradation in supercritical flow with the purpose of investigating its propagations scales as, to the best of our knowledge, prior investigations have been conducted only for subcritical conditions. The study is articulated around the following questions: (1) How can one quantify the celerity of propagation of an aggradation wave? (2) How does the aggradation wave celerity correlate with the celerity of small perturbations determined by the eigenvalues of the governing equations? (3) Which is the impact of considering or discarding the sediment concentration on the previous point? These issues are investigated with reference to a prototypal aggradation experiment performed in a laboratory flume. The manuscript is structured as follows: Section 2 provides a review of how the eigenvalues of the hyperbolic equations have been treated by several studies with or without an assumption of negligible sediment concentration. Section 3 borrows from hydro-dynamics a method to estimate the local and instantaneous celerity of propagation of an aggradation wave. In section 4, the experimental setup and the measurement techniques are presented; furthermore, the section describes the methods for the quantitative determination of the different celerities. Sections 5 and 6 contain the experimental results and a discussion, respectively. Finally, the key conclusions of the study are provided.

## 2 Mathematical modeling of channel morphologic evolution: a review of quasi-two-phase approaches and the celerity of small perturbations

In a fully-two-phase approach, the governing equations for one-dimensional modeling of channel morphologic evolution comprehend four partial differential equations (e.g., Greco et al., 2012), including two mass and momentum equations for the mixture phase and the same equations for the solid phase. However, since in the literature a quasi-two-phase approach has been widely taken to simulate mobile-bed flows, we follow this approach also in this study. A quasi-two-phase model for river morphology is based on two main hypotheses (e.g., Garegnani et al., 2013). First, if the volumetric concentration of the transported sediment, $c_s$, remains below a certain threshold, then the bed shear stress may be computed using the same equations one uses for clear-water conditions. For example, Armanini et al. (2009) proposed a threshold value of 0.05. The second hypothesis is that water and sediment move at almost the same velocity. In a quasi-two-phase approach, a hyperbolic system of partial differential equations includes one continuity equation for the mixture, one momentum equation for the

mixture, and one continuity equation for the bed sediment (Exner equation). The two equations for the mixture can be simplified to the well-known Saint-Venant equations for clear water if the solid concentration is negligible. The next sub-sections separately review the cases for negligible and non-negligible sediment concentration.

## 2.1 Governing equations and eigenvalue analysis for negligible $c_s$

Several authors (e.g., De Vries, 1965, 1971, 1973, 1993; Sieben, 1997, 1999; Armanini, 2018; Goutière et al., 2008; Garegnani et al., 2011, 2013) have argued that in fluvial environments the volumetric solid concentration may be assumed to be very small. With this assumption, the Saint-Venant and Exner equations can be formulated as follows:

$$\begin{cases} \dfrac{\partial h}{\partial t} + \dfrac{\partial (uh)}{\partial x} = 0 \\ \dfrac{\partial (uh)}{\partial t} + \dfrac{\partial (u^2 h)}{\partial x} + gh\left(\dfrac{\partial h}{\partial x} + \dfrac{\partial z_b}{\partial x}\right) = -ghS_f \\ (1 - p_0)\dfrac{\partial z_b}{\partial t} + \dfrac{\partial q_s}{\partial x} = 0 \end{cases} \tag{1}$$

where $h$ = water depth; $u$ = depth-averaged water velocity; $g$ = gravity acceleration; $z_b$ = bed elevation; $S_f$ = friction slope; $p_0$ = bed porosity; and $q_s$ = sediment discharge per unit width. The system (1) is a simplified form of the Saint-Venant and Exner equations considering unit width of a large channel, where the sectional area can be approximated as the product of a width and the flow depth. This formulation avoids including the flow area and discharge in the equations, and is typically used to write the system in vector form and determine its eigenvalues, as will be described below. This system contains five unknowns, namely $h, u, z_b, S_f, q_s$, and only three equations; therefore, in order to ensure the existence of a solution two other equations, working as closure relationships, are needed. The latter express $S_f$ and $q_s$. For the friction slope, the Manning formula is frequently used:

$$S_f = \frac{n^2 u^2}{R_H^{4/3}} \tag{2}$$

where $n$ = Manning's coefficient; and $R_H$ = hydraulic radius. To evaluate the sediment discharge, $q_s$, many formulae are available. For example, Goutiére et al. (2008) expanded the formula of Meyer-Peter and Müller (1948) as follows:

$$q_s(q, h) = 8\sqrt{g(s - 1)d_{50}^3}\left(\frac{n^2 q^2}{(s - 1)d_{50}h^{7/3}} - 0.047\right)^{3/2} \tag{3}$$

where $q = uh$ = water discharge per unit width; $s = \rho_s/\rho$ = relative sediment density ($\rho_s$ and $\rho$ represent the sediment and water densities, respectively); and $d_{50}$ = median grain diameter.

Exploiting a compound derivative for $\partial q_s/\partial x$ and the derivation of the product for $\partial(uh)/\partial x$, system (1) becomes:

$$\begin{cases} \dfrac{\partial h}{\partial t} + u\dfrac{\partial h}{\partial x} + h\dfrac{\partial u}{\partial x} = 0 \\ \dfrac{\partial u}{\partial t} + g\dfrac{\partial h}{\partial x} + u\dfrac{\partial u}{\partial x} + g\dfrac{\partial z_b}{\partial x} = -gS_f \\ \dfrac{\partial z_b}{\partial t} + \dfrac{1}{(1 - p_0)}\dfrac{\partial q_s}{\partial h}\dfrac{\partial h}{\partial x} + \dfrac{1}{(1 - p_0)}\dfrac{\partial q_s}{\partial u}\dfrac{\partial u}{\partial x} = 0 \end{cases} \tag{4}$$

System (4) can be written in the vector form as (Armanini, 2018):

$$\frac{\partial \boldsymbol{U}}{\partial \boldsymbol{t}} + \boldsymbol{A_U}\frac{\partial \boldsymbol{U}}{\partial \boldsymbol{x}} + \boldsymbol{S_U} = 0 \tag{5}$$

where:

$$\boldsymbol{U} = \begin{bmatrix} h \\ u \\ z_b \end{bmatrix} \qquad \boldsymbol{A_U} = \begin{bmatrix} u & h & 0 \\ g & u & g \\ \frac{1}{(1-p_0)}\frac{\partial q_s}{\partial h} & \frac{1}{(1-p_0)}\frac{\partial q_s}{\partial u} & 0 \end{bmatrix} \qquad \boldsymbol{S_U} = \begin{bmatrix} 0 \\ gS_f \\ 0 \end{bmatrix} \tag{6}$$

are the vector of unknowns, the matrix of coefficients, and the vector of known terms, respectively.

The eigenvalues ($\lambda_1, \lambda_2$, and $\lambda_3$) of the coefficient matrix $\boldsymbol{A_U}$ represent the slope of the so-called characteristic lines. The latter are described by the equation $\lambda_i = dx/dt$ for $i$=1,3; along these lines discontinuities, such as infinitesimal perturbations in the system variables, can propagate (Armanini, 2018). Lyn and Altinakar (2002) discussed that, after linearizing the system, the eigenvectors of the system determine the Riemann invariants. They interpreted the Riemann invariants as quantities that travel

along the characteristic lines at speeds equal to the eigenvalues. Building on these insights, several studies (e.g., De Vries, 1965, 1971, 1973; Lyn, 1987; Lyn and Altinakar, 2002; Goutière et al., 2008; Armanini, 2018) have treated the eigenvalues of the system as the celerity of propagation of small perturbations in the bed and water surfaces. For the more simple case of water ripples, mentioned in the Introduction, it can be demonstrated that the eigenvalues of the governing system of equations, $u \pm \sqrt{gh}$, indeed coincide with the celerity one obtains imposing the conservation of mass and energy (e.g., Chanson, 2004,

p. 225; Çengel and Cimbala, 2006, p. 685; Munson et al., 2013, p. 556).

The three eigenvalues of the system can be obtained by solving the following equation:

$$det(\boldsymbol{A_U} - \lambda \boldsymbol{I}) = \boldsymbol{0} \tag{7}$$

where $\boldsymbol{I}$ is the identity matrix. By developing (7) one obtains a cubic equation which is known as the characteristic polynomial equation (Armanini, 2018):

$$p(\lambda) = -\lambda^3 + 2u\lambda^2 + (gh - u^2 + ghA_\lambda)\lambda - ghu(A_\lambda - B_\lambda) = 0 \tag{8}$$

where:

$$A_\lambda = \frac{1}{(1-p_0)h}\frac{\partial q_s}{\partial u} \quad , \quad B_\lambda = \frac{1}{(1-p_0)u}\frac{\partial q_s}{\partial h} \tag{9}$$

are dimensionless parameters. Though the three real and distinct eigenvalues of the system can be computed exactly by solving this cubic equation (e.g., Rosatti and Fraccarollo, 2006), approximated solutions may be useful for interpretation purposes (Lyn, 1987). Numerous studies have thus been performed to investigate the characteristic lines (Lyn and Altinakar, 2002) and present approximated formulations for $\lambda_1, \lambda_2$, and $\lambda_3$. In the following, some of them are reviewed. It is noted that the approximated solutions may sometimes yield non-real roots, depending on the Froude number ($Fr = u/\sqrt{gh}$).

**2.1.1 De Vries' approach**

De Vries (1965, 1971, 1973, 1993) referred to the eigenvalues of the system as the "celerities of the surface and bed waves in mobile-bed flows". To estimate them, he proposed an approximated solution valid for Froude numbers lower than 0.8 or higher than 1.2:

$$\begin{cases} \lambda_1 \cong \left[1 + \frac{1}{Fr}\right]u \\ \lambda_2 \cong \left[1 - \frac{1}{Fr}\right]u \\ \lambda_3 \cong \frac{u}{1 - Fr^2}(A_\lambda - B_\lambda) \end{cases} \tag{10}$$

In Eq. (10), one recognizes that $\lambda_1$ and $\lambda_2$ are the well-known surface wave celerities for the Saint-Venant equations

representing water flow. In this approach, therefore, for a flow sufficiently distant from the critical conditions and with

negligible solid concentration, the first two eigenvalues are not affected by the presence of sediment transport. Instead, $\lambda_3$ was considered as the celerity of the bed surface perturbations.

The eigenvalues $\lambda_2$ and $\lambda_3$ have a different sign in subcritical and supercritical flow. Therefore, an alternative version of Eq. (10) was proposed to obtain eigenvalues with the same sign ($\lambda_1 > 0$, $\lambda_2 < 0$, $\lambda_3 > 0$) for both the flow conditions:

$$Fr < 0.8 \begin{cases} \lambda_1 \cong \left[1 + \dfrac{1}{Fr}\right]u \\[2mm] \lambda_2 \cong \left[1 - \dfrac{1}{Fr}\right]u \\[2mm] \lambda_3 \cong \dfrac{u}{1 - Fr^2}(A_\lambda - B_\lambda) \end{cases} \qquad Fr > 1.2 \begin{cases} \lambda_1 \cong \left[1 + \dfrac{1}{Fr}\right]u \\[2mm] \lambda_2 \cong \dfrac{u}{1 - Fr^2}(A_\lambda - B_\lambda) \\[2mm] \lambda_3 \cong \left[1 - \dfrac{1}{Fr}\right]u \end{cases} \qquad (11)$$

### 170  2.1.2 Lyn and Altinakar's approach

Lyn (1987), then followed by Lyn and Altinakar (2002), presented different approximations to estimate the three eigenvalues for near-critical flows ($0.8 \leq Fr^2 \leq 1.2$):

$$\begin{cases} \lambda_1 \cong \left[\dfrac{3}{2} + \dfrac{1}{2Fr}\right]u \\[3mm] \lambda_2 \cong \left[\dfrac{1}{4}\left(1 - \dfrac{1}{Fr^2}\right) - \dfrac{1}{4}\sqrt{\left(1 - \dfrac{1}{Fr^2}\right)^2 + \dfrac{8A_\lambda}{Fr^2}}\right]u \\[3mm] \lambda_3 \cong \left[\dfrac{1}{4}\left(1 - \dfrac{1}{Fr^2}\right) + \dfrac{1}{4}\sqrt{\left(1 - \dfrac{1}{Fr^2}\right)^2 + \dfrac{8A_\lambda}{Fr^2}}\right]u \end{cases} \qquad (12)$$

In agreement with Sieben (1997, 1999), Lyn and Altinakar (2002) argued that in near-critical regimes, bed waves interact strongly with surface waves, and $\lambda_2$ and $\lambda_3$ are not devoted merely to a surface wave or a bed wave but, rather, they represent
### 175  the celerity of propagation of both surface and bed waves.

### 2.1.3 Goutière et al.'s approach

Goutière et al. (2008) developed approximated formulations for the eigenvalues of a system of partial differential equations slightly different from (1), where $h$, $q$ and $z_b$ were the main dependent variables:

$$\begin{cases} \dfrac{\partial h}{\partial t} + \dfrac{\partial q}{\partial x} = 0 \\[3mm] \dfrac{\partial q}{\partial t} + \dfrac{\partial}{\partial x}\left(\dfrac{q^2}{h}\right) + gh\left(\dfrac{\partial h}{\partial x} + \dfrac{\partial z_b}{\partial x}\right) = -ghS_f \\[3mm] (1 - p_0)\dfrac{\partial z_b}{\partial t} + \dfrac{\partial q_s}{\partial x} = 0 \end{cases} \qquad (13)$$

To estimate the three eigenvalues of the system they proposed the following equations:

$$\begin{cases} \lambda_1 \cong \left[1 + \dfrac{1}{Fr}\right]u \\[3mm] \lambda_2 \cong \dfrac{1}{2}\left[\left(1 - \dfrac{1}{Fr}\right) - \sqrt{\left(1 - \dfrac{1}{Fr}\right)^2 - \dfrac{4(A_\lambda - B_\lambda)}{(Fr^2 + Fr)}}\right]u \\[3mm] \lambda_3 \cong \dfrac{1}{2}\left[\left(1 - \dfrac{1}{Fr}\right) + \sqrt{\left(1 - \dfrac{1}{Fr}\right)^2 - \dfrac{4(A_\lambda - B_\lambda)}{(Fr^2 + F_r)}}\right]u \end{cases} \qquad (14)$$

### 180  Differently from the approaches of De Vries and Lyn and Altinakar, Eq. (14) are to be intended as valid for the whole range of Froude numbers.

**2.2 Governing equations and eigenvalue analysis for non-negligible $c_s$**

Morris and Williams (1996) argued that an assumption of negligible solid concentration is not appropriate for many natural streams and, therefore, determined the eigenvalues of a system of equations considering a finite $c_s$. The continuity and momentum equations of the mixture and the continuity equation for the sediment in Morris and Williams' approach are as follows:

$$
\begin{cases}
\dfrac{\partial (uh)}{\partial x} + \dfrac{\partial h}{\partial t} + \dfrac{\partial z_b}{\partial t} = 0 \\[2mm]
\dfrac{\partial u}{\partial t} + u\dfrac{\partial u}{\partial x} + g\dfrac{\partial h}{\partial x} + \dfrac{(\rho_s - \rho)gh}{2\rho_m}\dfrac{\partial c_s}{\partial x} - \dfrac{[(1-p_0)\rho_s + p_0\rho]u}{\rho_m h}\dfrac{\partial z_b}{\partial t} + g\dfrac{\partial z_b}{\partial x} = -gS_f \\[2mm]
\dfrac{\partial (uhc_s)}{\partial x} + \dfrac{\partial (hc_s)}{\partial t} + (1-p_0)\dfrac{\partial z_b}{\partial t} = 0
\end{cases}
\tag{15}
$$

where, $c_s = q_s/(q_s + q)$, and $\rho_m = c_s\rho_s + (1-c_s)\rho$ is the density of the mixture. Since the solid concentration is not negligible, the water and sediment discharge per unit width are obtained from the following equations (always assuming that the solid particles move with the same velocity of water):

$$q = uh(1 - c_s) \tag{16}$$

$$q_s = uhc_s \tag{17}$$

The system can be again closed using Eq. (2) and Eq. (3), as the latter furnishes a mean to compute $c_s$ from Eq. (17). The eigenvalues of the system (15) are determined by solving the following cubic equation:

$$
\lambda^3 \left\{ Bu\frac{\partial c_s}{\partial u} - h\frac{\partial c_s}{\partial h} - [c_s - (1-p_0)] \right\} + \lambda^2 \left( \{Agh[c_s - (1-p_0)] - 2Bu^2\}\frac{\partial c_s}{\partial u} + (2+B)uh\frac{\partial c_s}{\partial h} + \right.
$$

$$
\left. 2u[c_s - (1-p_0)] \right) + \lambda\left[ (Bu^3 - ugh\{1 + A[c_s - (1-p_0)]\})\frac{\partial c_s}{\partial u} - ((1+B)u^2h - gh^2\{1 + \right.
$$

$$
\left. A[c_s - (1-p_0)]\})\frac{\partial c_s}{\partial h} - (u^2 - gh)[c_s - (1-p_0)] \right] + ugh\left( u\frac{\partial c_s}{\partial u} - h\frac{\partial c_s}{\partial h} \right) = 0
\tag{18}
$$

where $A = (\rho_s - \rho)/(2\rho_m)$ and $B = ((1-p_0)\rho_s + p_0\rho)/\rho_m$ are dimensionless parameters. Differently from the previous approaches for negligible $c_s$, this last one does not carry explicit approximated equations for the $\lambda$ values. For the sake of completeness, an extended report of Morris and Williams' derivations is included in Supplemental 1.

**3 Estimation of the local and instantaneous celerity of propagation of sediment aggradation**

To determine the celerity of propagation of a certain quantity $X$, that varies in space and time, we borrow the developments provided for unsteady flow by, for example, Chow et al. (1988, p. 284) or Jain (2001, p. 240). The celerity of propagation is defined as the velocity at which a given value of that quantity migrates along the system with respect to a still observer or, conversely, the velocity at which an observer needs to move to see a constant value for the quantity. According to this definition, the celerity of propagation of $X$, $C_X$, can be obtained by imposing that the total differential of the quantity is zero over an infinitesimal time interval, which corresponds to $dX/dt = 0$:

$$
\frac{dX}{dt} = \frac{\partial X}{\partial t} + \frac{\partial X}{\partial x}\frac{dx}{dt} = 0 \;\rightarrow\; C_X = \frac{dx}{dt} = -\frac{\partial X/\partial t}{\partial X/\partial x}
\tag{19}
$$

In a clear-water wave model, suitable quantities to be used as $X$ are, for example, water depth or discharge; in a kinematic-wave model the celerities of these two quantities are equal, while in a diffusive wave they are different (see again Chow et al.'s and Jain's books). In the present case, any quantity related to the morphologic process may be used; in the next section we will choose to use the bed elevation.

## 4 Materials and methods

The aggradation experiment presented in this paper was conducted at the Mountain Hydraulics Laboratory of the Politecnico di Milano, Lecco campus (Italy). It was one of twenty-five experiments within an experimental campaign, and it is used as a proof of concept in the manuscript. Supplemental 2 provides experimental parameters and results for the others, as a support of the interpretation and conclusions drawn in the manuscript. The experimental setup and procedures are briefly reported below, referring to prior work for details.

### 4.1 Experimental setup, procedures, and measuring methods

The configuration of the used flume is shown in Fig. 2. The flume has a length of 5.2 m, a width of 0.3 m, a bank height of 0.45 m, and is entirely made of Plexiglas. The water discharge, $Q$, pumped from an underground container into an upstream tank, is adjusted using a guillotine valve and measured by an electromagnetic flowmeter. The erodible channel bed has a thickness of 0.15 m and is made of Polyvinyl Chloride (PVC), uniform sediment cylinders with a density $\rho_s$ of 1443 kg/m³, an equivalent diameter (computed for a sphere with the same volume of a cylinder) $d$ of 3.8 mm, porosity $p_0$ of 0.45 (Unigarro Villota, 2017), and a Manning roughness coefficient $n$ of 0.015 s/m$^{1/3}$ (as determined during preliminary tests by Zanchi and Radice, 2021). Sediment is fed at 30 cm from the inlet by a system with a vibrating channel and a hopper. At the downstream end, the channel is equipped with sediment collectors. The bed is fixed in an upstream and a downstream portion to avoid undesired scour. A laser distance sensor is installed to continuously measure the water elevation in the inlet tank.

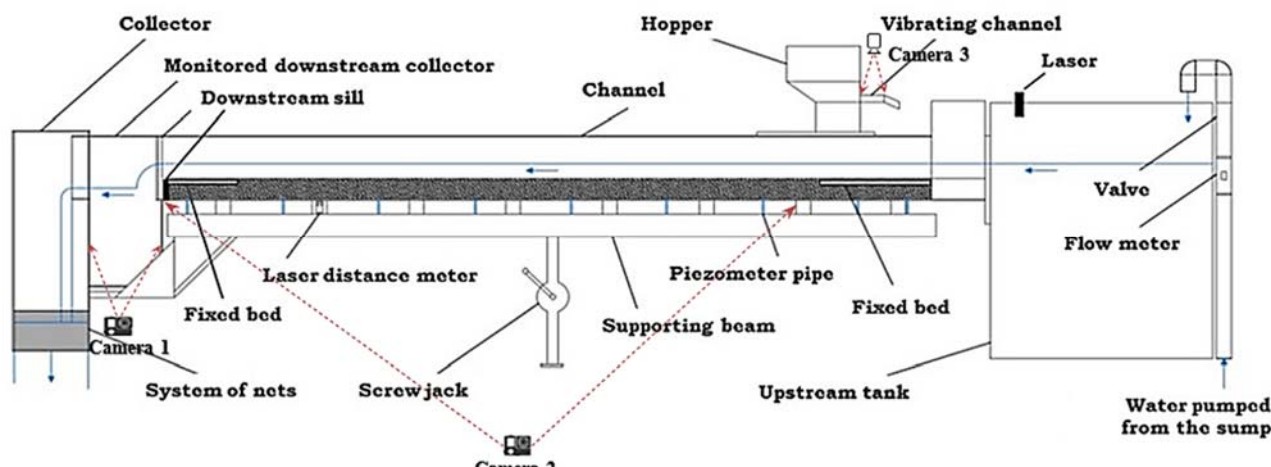

**Figure 2. Flume used for the aggradation experiment. Camera 1 shoots the collector; camera 2 shoots the channel from the side; camera 3 shoots the vibrating channel of the sediment feeding system.**

Prior to executing an experiment, the sediment bed is leveled and sprayed with water to avoid surface tension effects at water arrival. Then, the water pumping system is switched on and the flume is filled at negligible discharge to avoid any significant disturbance to the sediment. In this term, several cameras placed around the flume (see again Fig. 2) are turned on. After saturation of the bed, which typically takes some minutes, the water discharge is increased to the test value in roughly one minute; in this phase, the continuous reading of the discharge measured by the flowmeter and the laser measurement of the water elevation in the inlet tank furnish the necessary data to compute the temporal evolution of the flow rate entering the flume. When the sediment in the flume starts being transported, the sediment feeding is activated (the time at which the feeder is turned on is considered the initial time of the experiment). During the experiment, the hopper is refilled manually, and the experiment ends when no more sediment material is available (the duration of the aggradation phase is generally less than 10 minutes).

All the measurements are taken by image analysis, using the videos acquired by the various cameras. First, the sediment feeding rate, $Q_{s-in}$, is obtained from the videos recorded by camera 3 placed above the vibrating channel. Particle image velocimetry is used to measure the velocity of the particles moving along the vibrating channel; then, this velocity is converted

into a sediment feeding rate using a predetermined transfer function (for details on the calibration of the method the reader can refer to Radice and Zanchi, 2018).

Second, in order to measure the bed and water elevation during the experiment, a motion-detection method is applied to the videos captured by camera 2 placed beside the channel. The camera furnishes videos of a portion of the flume, starting at 140 cm from the inlet, to the downstream end. For the measurement of the bed elevation, this method is based on the concept that the bed-load layer is part of the mixture flow while the bed is atop the still sediment in the bottom layer; according to that, the bed is located at the maximum elevation where no sediment motion occurs (see Eslami et al., 2022). For the measurement of

the water surface elevation, instead, the method relies on detecting the motion of sediment particles sparsely transported within the flow. Two subsequent frames of the movie are subtracted from each other and imposing a threshold value to the resulting difference determines the border between motion and stillness; the upper and lower edges of the detected motion layer are considered as the water and bed surface elevations, respectively. To validate the method, manual measurements were also taken during the experiment. Before starting the experiment, three rulers were installed at the upstream, middle, and

downstream sections of the flume. Three observers were positioned at these locations, ensuring they avoided interfering with the cameras' field of view. During the experiment, the observers recorded the water and bed surface levels at various time instants using the rulers. At the end of the experiment, the results obtained from the automatic bed and water detection method were compared with the manually recorded values at corresponding times and locations. The comparisons showed satisfactory agreement, with mean squared errors of 4.0 mm² for the bed surface and 1.5 mm² for the water surface.

Third, the sediment transport capacity of the initial flow, $Q_{s0}$, is measured by two methods, one relying on the sediment volume progressively accumulated into the downstream collector and one obtaining the sediment volume leaving the flume from those fed and accumulated in the channel (thus applying the mass conservation principle). Full details are also provided in Eslami et al. (2022). The measurement of $Q_{s0}$ is necessary to obtain the sediment overloading ratio $Lr = Q_{s-in}/Q_{s0}$.

**4.2 Aggradation experiment and raw results**

Table 1 lists the control parameters of the aggradation experiment. Symbols not already defined in the text are $T$, representing the experiment duration; $S_0$, the initial slope of the channel; $Q$, the water discharge. Furthermore, $H$ and $U$ denote water depth and velocity, respectively, calculated using the Gauckler-Strickler formula for uniform flow with the initial slope. While water depth and velocity continuously vary in space and time as aggradation proceeds, these parameters are used as reference values for the initial condition of the experiment. Consequently, for example, the Froude number in this table is calculated as $Fr = $

$U/\sqrt{gH}$, and the Reynolds number is calculated as $Re = \rho UH/\mu$, where $\rho$ and $\mu$ are density and dynamic viscosity of water.

**Table 1. Parameters of the aggradation experiment performed in the present study.**

| $T$ (s) | $S_0$ (%) | $Q$ ($m^3/s$) | $Q_{s-in}$ ($m^3/s$) | $Q_{s0}$ ($m^3/s$) | $Lr$ | $H$ ($m$) | $U$ ($m/s$) | $Fr$ | $Re$ |
|---|---|---|---|---|---|---|---|---|---|
| 316 | 1.37 | 0.007 | 4.28E-04 | 1.33E-04 | 3.22 | 0.033 | 0.705 | 1.236 | 19113 |

The raw results of the experiment are presented in this "method" section to serve as a basis for describing the other methods used in the analysis. During the experiment, the bed elevation progressively increased over time (Fig. 3), with a more

pronounced rise upstream than downstream, causing the slope to increase; the water profile changed accordingly. The slope adjustment enhanced the sediment transport rate; the latter tended to reach the sediment feeding rate, leading the channel towards an equilibrium condition. The elevation rose rapidly at the beginning of the experiment, but gradually slowed for larger time. The channel reached an equilibrium state around 190 seconds, after which the profiles showed no further evolution. A sediment front was not clearly detectable from the bed surface profiles during the experiment, due to dispersive aggradation

induced by a relatively high flow velocity, as reported in Table 1.

(a)

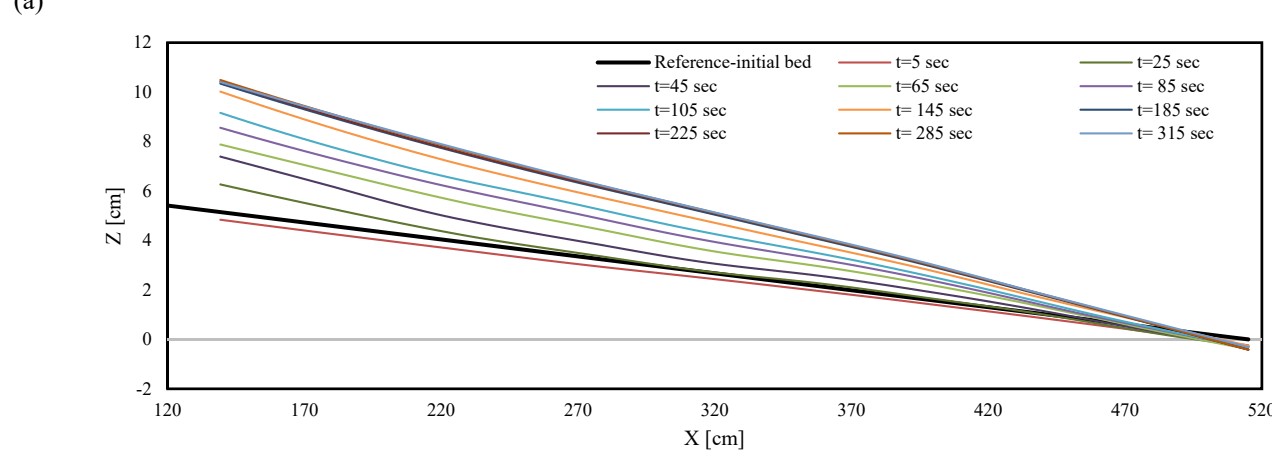

(b)

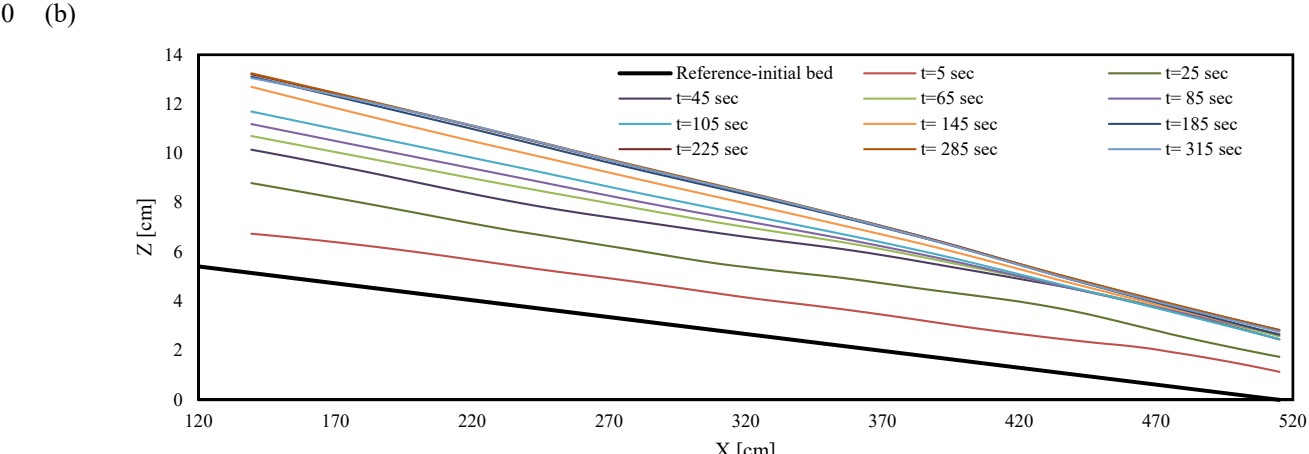

**Figure 3. Spatial profiles of (a) bed elevation (b) water elevation, during the experiment.**

The spatial and temporal evolutions can be observed simultaneously by depicting color gradient maps for the elevations of the bed and water surface (Fig. 4). The contour lines become vertical for times larger than 190 s, again demonstrating the

285 achievement of morphological equilibrium in the performed experiment. Finally, a color gradient map (Fig. 5) also shows that the Froude number, obtained from local and instantaneous velocity and depth, was larger than 1 for most of the experiment (with a mean value of 1.3 for the entire map), corresponding to a supercritical flow condition.

(a)                                                          (b)

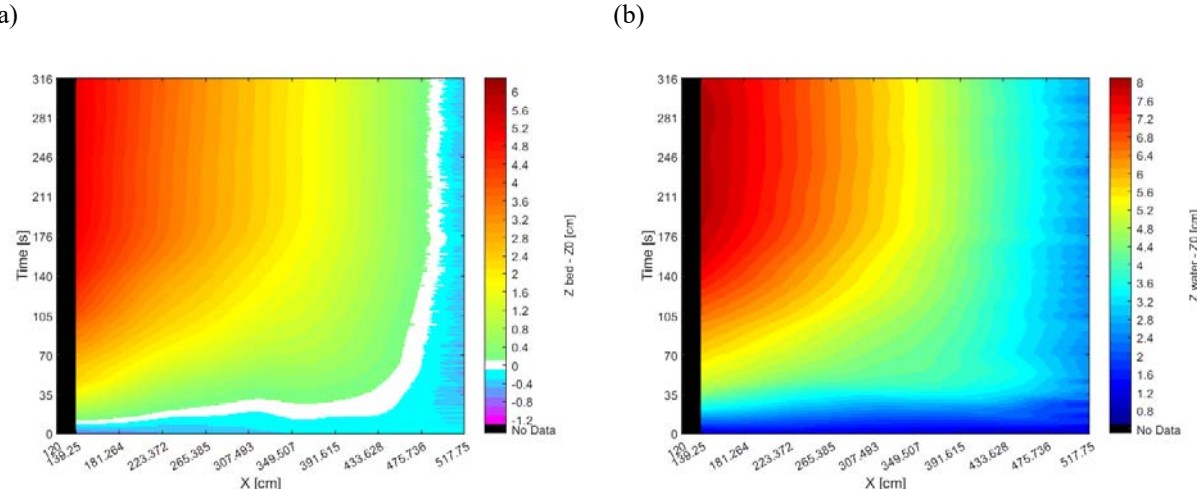

**Figure 4. Color gradient maps of (a) bed elevation (b) water elevation, for the performed experiment.**

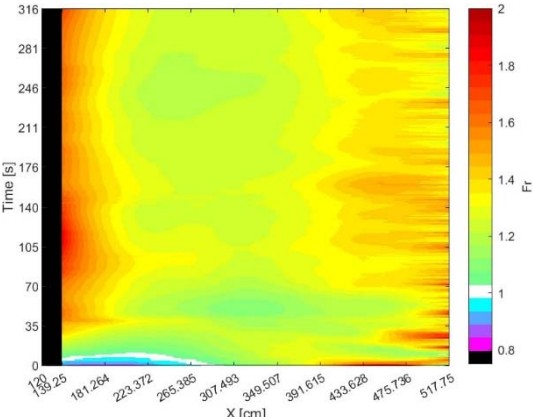

Figure 5. Color gradient map of Froude number for the performed experiment.

### 4.3 Determination of the local and instantaneous celerity of propagation of sediment aggradation

Eq. (19) is applied to the bed elevation $z_b$ that is the most suitable quantity to determine the celerity of the sediment wave. The bed elevation was preferred to, for example, the sediment transport rate because it was directly measured during the experiment with high spatial and temporal resolution. To exploit the equation for the present purposes, it was converted into a discrete form:

$$C = -\frac{\Delta z_b/\Delta t}{\Delta z_b/\Delta x} \tag{20}$$

where $\Delta t$ and $\Delta x$ are suitable time and space steps. Due to the swiftness of the aggradation process under supercritical flows, and thus to ensure the availability of sufficient data for an accurate estimation of $C$ in terms of time, $\Delta t$ was set to 1 second. This choice aligns with the temporal resolution of the measurement method used to capture bed and water surface elevations. For $\Delta x$, a trade-off between computational cost and estimation accuracy was determined through a trial-and-error approach, leading to the selection of $\Delta x$=1.8 cm. The determinations of celerity by (20) were compared to alternative ones picking a certain value of bed elevation in the bed profile for a certain time, and looking for the same value in the bed profile for a following time; the ratio of a determined travelled distance to the time interval separating the two profiles yielded a value of $C$. The exercise confirmed the expected suitability of eq. (20) for the determination of the local and instantaneous celerity of the aggradation wave. In the context of the initial questions of the paper, the use of eq. (20) thus constitutes our answer to question 1.

### 4.4 Eigenvalue determination using approaches for negligible and non-negligible sediment concentration

Within the approaches for negligible solid concentration, the equations proposed by Goutière et al. (2008) are used in this study, since these equations are valid for the entire range of the Froude number providing determinations very similar to those from the exact solution of Eq. (8) with three real roots. Since all the quantities involved in the eigenvalue determination vary in space and time, one obtains a corresponding space-time variation of the $\lambda$ values. In order to make the eigenvalue determination fully consistent with the performed experiment, Eq. (3) was preliminarily calibrated introducing a bed-load factor $\alpha$ and an equivalent Manning coefficient accounting for sediment transport (excluding for simplicity the threshold Shields number from the calibration parameters):

$$q_s(q,h) = \alpha 8\sqrt{g(s-1)d_{50}^3}\left(\frac{n_{calibrated}^2 q^2}{(s-1)d_{50}h^{7/3}} - 0.047\right)^{3/2} \tag{21}$$

Following Eslami et al. (2022), the calibration was performed by using Eq. (21) in a numerical simulation of the experiment and achieving a good consistency between experimental and numerical profiles of the bed and water surface. The values

obtained for $\alpha$ and $n_{calibrated}$ were equal to 1.73 and 0.017 s/m$^{1/3}$, respectively. Furthermore, by substituting $q = uh$ in Eq. (21), explicit equations for $\partial q_s/\partial u$ and $\partial q_s/\partial h$ were obtained to determine $A_\lambda$ and $B_\lambda$ from Eq. (10). Finally, $\lambda_1, \lambda_2$, and $\lambda_3$ could be computed by Eq. (14).

There is no explicit equation in the literature to estimate $\lambda_1, \lambda_2$, and $\lambda_3$ for non-negligible solid concentration; therefore, in the current study, the eigenvalues were determined by finding the roots of Eq. (18). Furthermore, since the sediment transport rate was expressed by Eq. (21), an equation for sediment concentration was obtained from that one as:

$$c_s(u, h) = \frac{8\alpha\sqrt{g(s-1)d_{50}^3}\left(\frac{n_{calibrated}^2(uh(1-c_s))^2}{(s-1)d_{50}h^{7/3}} - 0.047\right)^{3/2}}{uh} \tag{22}$$

to be solved iteratively to determine $c_s$. Additionally, Eq. (18) required the derivatives $\partial c_s/\partial u$ and $\partial c_s/\partial h$, that were obtained introducing a function $F(u, h, c_s)$ as follows:

$$F(u, h, c_s) = \frac{8\alpha\sqrt{g(s-1)d_{50}^3}\left(\frac{n_{calibrated}^2(uh(1-c_s))^2}{(s-1)d_{50}h^{7/3}} - 0.047\right)^{3/2}}{uh} - c_s \tag{23}$$

and, finally, determining the needed derivatives

$$\frac{\partial c_s}{\partial u} = -\frac{\partial F(u, h, c_s)/\partial u}{\partial F(u, h, c_s)/\partial c_s} \tag{24}$$

$$\frac{\partial c_s}{\partial h} = -\frac{\partial F(u, h, c_s)/\partial h}{\partial F(u, h, c_s)/\partial c_s} \tag{25}$$

## 5 Results

### 5.1 Local and instantaneous celerity of propagation of sediment aggradation

The color gradient map of $C$ as obtained by Eq. (20) is depicted in Fig. 6(a) with a dimensionless counterpart in Fig. 6(b). To provide the dimensionless map, the local and instantaneous flow velocity was used as a normalization parameter. The space-time distribution of the celerity values was smoothed by replacing any 64 values (8×8 values in space and time) with their average.

The local and instantaneous celerity tends to reach a zero value at the time around 190 s, coinciding with the previously mentioned achievement of a morphologic equilibrium. In fact, as the system tends to equilibrium, the temporal rate of bed elevation change diminishes, leading the propagation celerity to also approach zero. While the average values of the celerity and dimensionless celerity by considering the whole duration of the experiment are equal to 0.393 cm/s and around 0.006, respectively, they are equal to 0.642 cm/s and 0.010 considering only times lower than 190 s (in both cases, all the locations along the channel are considered).

(a) (b)

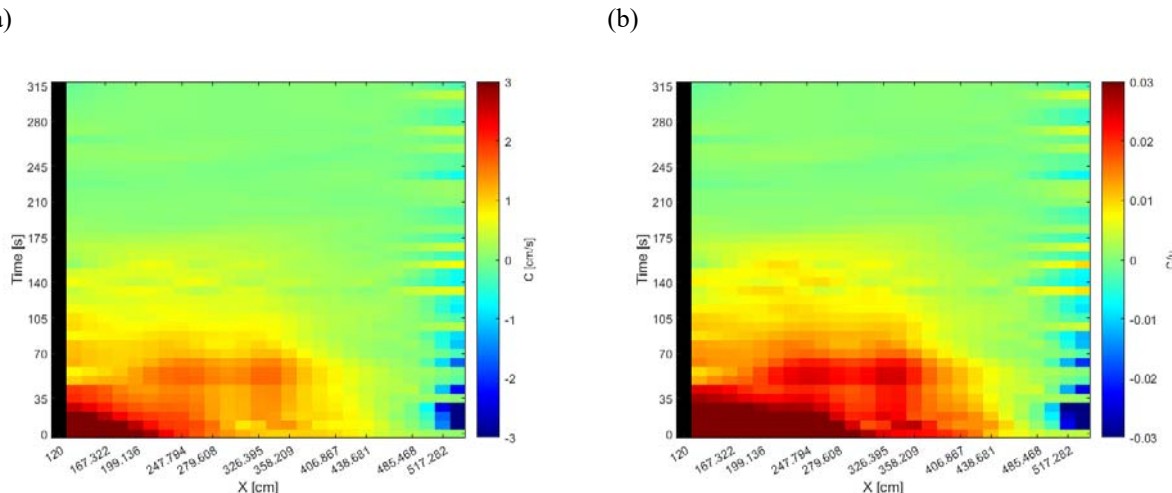

**Figure 6. Color gradient maps of (a) dimensional and (b) dimensionless local and instantaneous celerity of propagation of sediment aggradation.**

### 5.2 Eigenvalues of the system of PDEs

Figure 7(a, b, c) shows the color gradient maps of the dimensionless eigenvalues, $\lambda_1/u, \lambda_2/u,$ and $\lambda_3/u$, obtained from the approximated solutions proposed by Goutière et al. (2008) and from the exact solutions of the equation obtained by Morris and Williams (1996). The smoothing, described above for the aggradation wave celerity, was also applied to these celerities of small perturbations. The average values of $\lambda_1/u, \lambda_2/u,$ and $\lambda_3/u$ for the entire duration of the experiment and until the equilibrium time are equal to 1.77, -0.21 and 0.43 and to 1.77, -0.13 and 0.46, respectively, for the approaches of Goutière et

al. (2008) and Morris and Williams (1996). Therefore, as part of addressing the third study question concerning the impact of including or excluding solid concentration in the governing equations, one may say that the eigenvalues $\lambda_1$ and $\lambda_3$ show negligible differences, while a significant variation is obtained for $\lambda_2$.

(a)

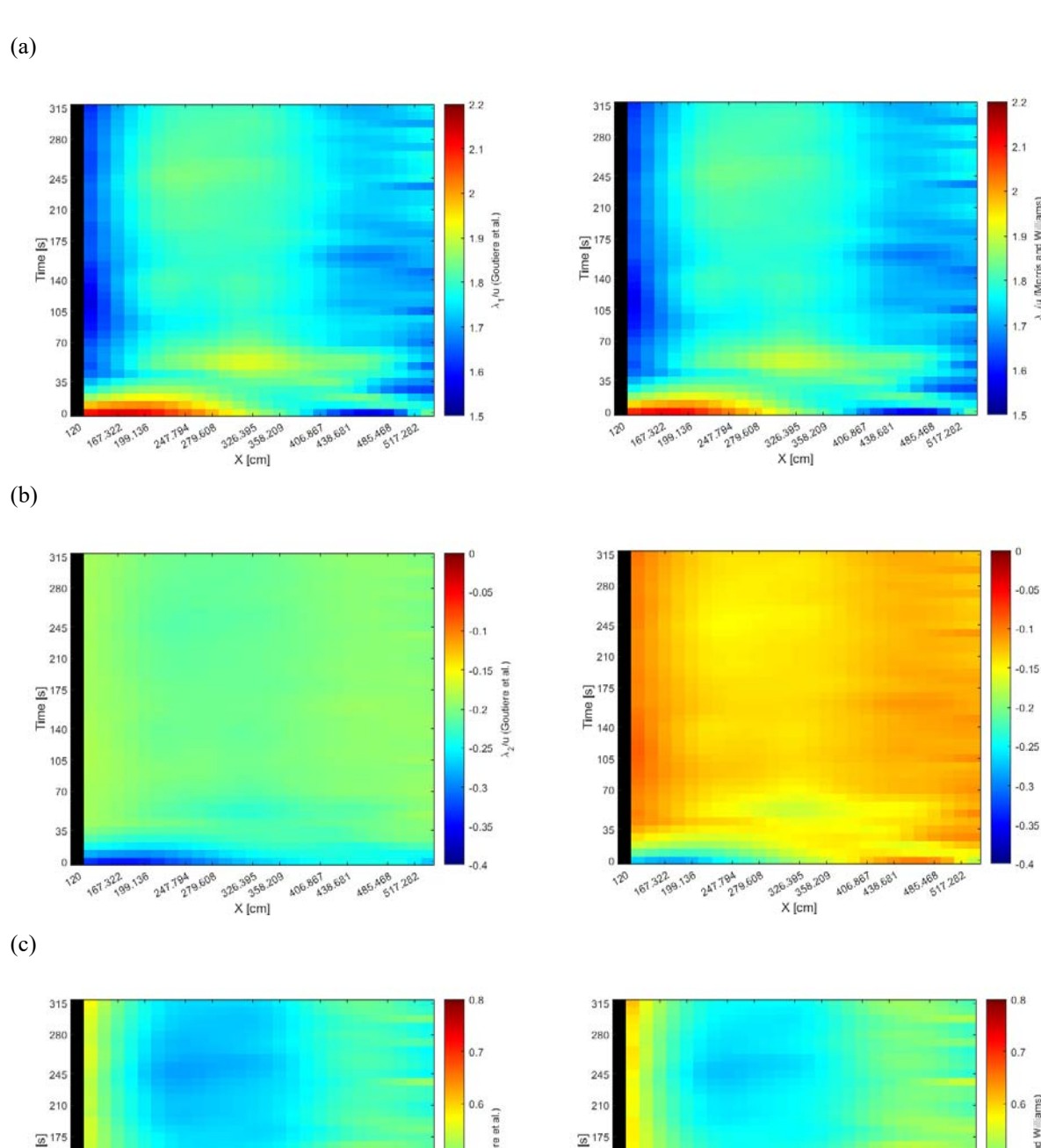

(b)

(c)

**Figure 7. Comparison between the color gradient maps of (a) $\lambda_1/u$, (b) $\lambda_2/u$ and (c) $\lambda_3/u$. The left graphs are obtained from the approximated solutions proposed by Goutière et al. (2008) and the right graphs are derived from the exact solution of Morris and Williams' (1996) approach.**

**5.3 Quantification of sediment concentration**

The smoothed color gradient map of the volumetric sediment concentration as obtained using Eq. (22) is depicted in Fig. 8. The average value of $c_s$ for the whole duration of the experiment and also until the equilibrium time is around 0.032, indicating that, according to the criteria proposed by De Vries (1965) with maximum $c_s$=0.002 and by Garegnani et al. (2011, 2013) with maximum $c_s$=0.01, the solid concentration was not negligible in the performed experiment. However, the average value of $c_s$ does not exceed the maximum value of 0.05 that was proposed by Armanini et al. (2009) for the validity of the quasi-two-phase approach.

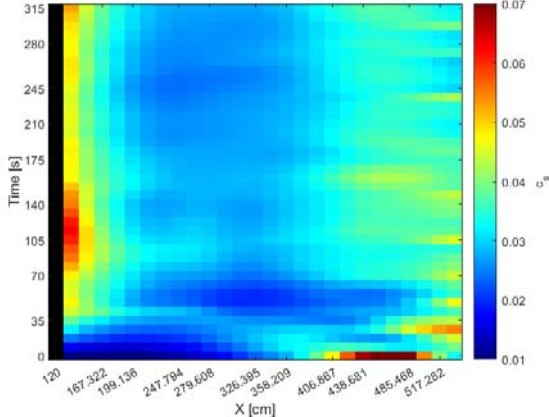

**Figure 8. Color gradient map of the volumetric solid concentration.**

## 5.4 Relationships between the two types of celerities and the Froude number

We first explore (Fig. 9) the relationship between the dimensionless local and instantaneous celerity of the aggradation wave and the Froude number during the experiment. For the sake of this comparison, the $Fr$ map of Fig. 5 was also smoothened as those of Fig. 6 and 7. Since the celerity tends to zero after the achievement of an equilibrium condition, only the points for $t <$ 190 s along the analyzed channel portion (e.g., 140 to 520 cm) are considered in this analysis. A general decrease of dimensionless celerity for increasing Froude number is observed. An opposite trend is seen for the highest points in the scatter, that are related to the initial stages of the experiment with a flow rate that was still increasing to the test value.

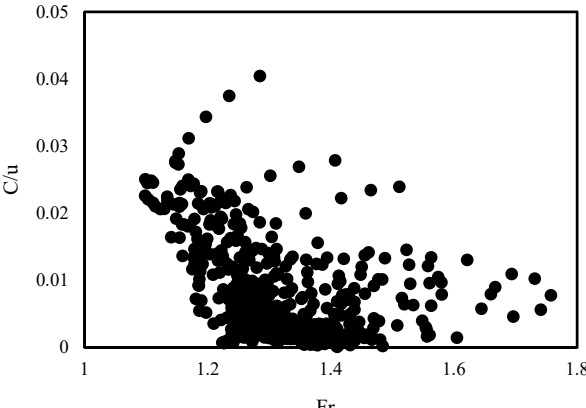

**Figure 9. Scatter plot showing the correlation between the Froude number and the local and instantaneous celerity of aggradation.**

The relationships between the Froude number and the dimensionless eigenvalues are presented in Fig. 10 for both the Goutière et al.'s and Morris and Williams' approaches. This depiction aligns with the typical curves presented in mathematical studies (e.g., Lyn and Altinakar, 2002, fig. 1; Garegnani et al., 2013, fig. 2), with the values that in the present paper were obtained from experimental parameters. The results further demonstrate the difference between the values obtained for $\lambda_2$ considering or discarding the sediment concentration.

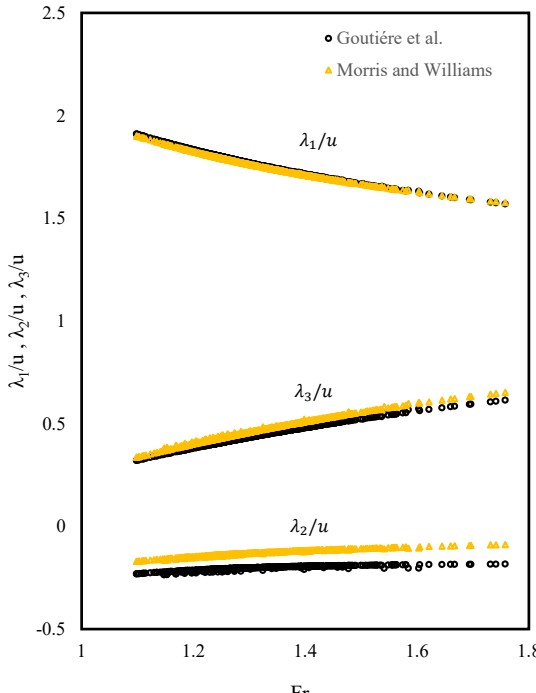

**Figure 10. Scatter plot showing the correlation between the Froude number and the dimensionless eigenvalues.**

(a)

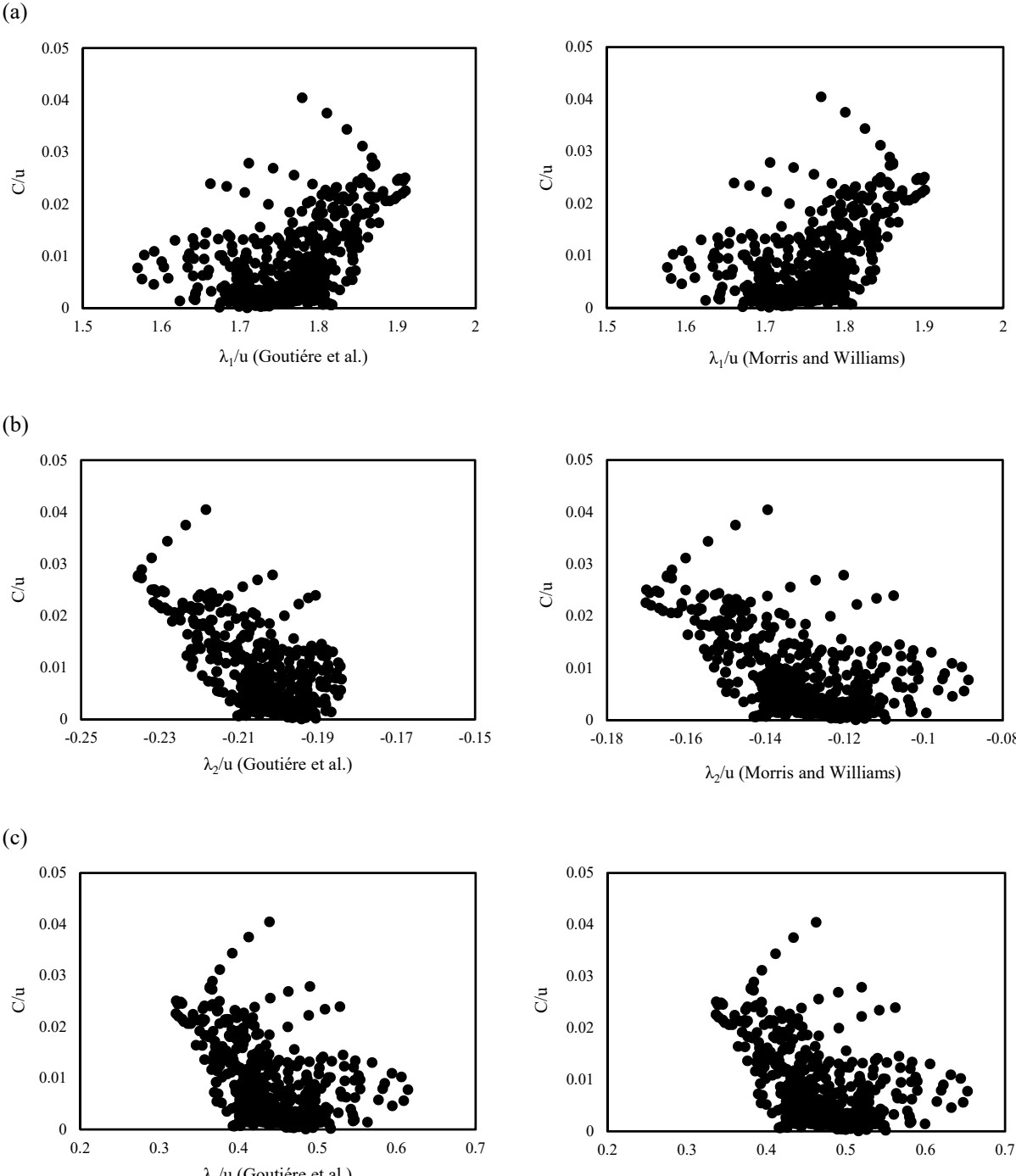

**Figure 11. Scatter plots showing the correlations between the dimensionless eigenvalues and celerity of the aggradation wave. The left graphs are obtained from the approximated solutions proposed by Goutière et al. (2008) and, the right graphs are derived from the exact solution of Morris and Williams' (1996) approach.**

Finally, Fig. 11(a, b, and c) shows how the dimensionless celerity of small perturbations and the dimensionless celerity of aggradation correlate with one another, again for both the Goutière et al.'s and Morris and Williams' approaches (left and right panels, respectively). These scatter plots address the second question of the present study, demonstrating that the dimensionless celerity of the aggradation wave increases with the dimensionless $\lambda_1$ and with the module of dimensionless $\lambda_2$ (that is negative), while it decreases with the dimensionless $\lambda_3$. Additionally, in response to the third question above, and complementing the findings in Sub-section 5.2, Fig. 11 highlights that considering or excluding sediment concentration in the governing equations does not alter the qualitative relationships between the different celerities. Only, Fig. 11(b) indicates that the variation of $\lambda_2$ is larger in the approach for non-negligible concentration than in the other one for negligible concentration. To illustrate the strength of the relationship between $C/u$ and $Fr$, as well as $C/u$ and $\lambda i/u$ (Fig. 9 and 11), the Pearson correlation

coefficient was calculated for the corresponding plots. Table 2 presents the correlation coefficients, which range between 0.4 and 0.6 for all plots, indicating a moderate correlation between the variables.

**Table 2. Pearson's correlation coefficient between the dimensionless local and instantaneous celerity of the aggradation wave and the Froude number, as well as the dimensionless eigenvalues, for the presented experiment.**

|  |  | Goutière et al. (2008) | | | Morris and Williams (1996) | | |
|---|---|---|---|---|---|---|---|
|  | $Fr$ | $\lambda_1/u$ | $\lambda_2/u$ | $\lambda_3/u$ | $\lambda_1/u$ | $\lambda_2/u$ | $\lambda_3/u$ |
| $C/u$ | -0.40 | 0.46 | -0.62 | -0.42 | 0.46 | -0.52 | -0.43 |

## 6 Discussion

### 6.1 Celerity determination

The measurement and analysis methods used in the present manuscript have enabled a depiction of the spatial and temporal evolutions of various quantities. In particular, based on the measured data, a local and instantaneous determination was achieved for the celerity of propagation of an aggradation wave and the eigenvalues of the equations modeling the process, which are representative of the celerity of small perturbations.

### 6.2 Relationships between the different celerities and the Froude number

We have used three types of scatter plots showing the correlation between different variables: $Fr - \lambda/u$ (Fig. 10), $Fr - C/u$ (Fig. 9), $\lambda/u - C/u$ (Fig. 11). The different trends of correlation are consistent with one another: for example, $Fr$ correlates negatively with both $\lambda_1/u$ and $C/u$, thus $\lambda_1/u$ and $C/u$ are positively correlated with each other. Similar cross-relationships are valid for the other eigenvalues and are reflected by the values of the Pearson coefficient in Table 2. The first type of correlation ($Fr - \lambda/u$) has been already presented and discussed in works related to the mathematical modelling of the hydro-morphologic process (e.g., De Vries, 1965, 1971, 1973, 1993; Lyn and Altinakar, 2002; Goutière et al., 2008; Armanini, 2018). Specifically, it has been shown that while $\lambda_1/u$ is a decreasing function of $Fr$, $|\lambda_2/u|$ and $\lambda_3/u$ increase with $Fr$, as also can be observed in Fig. 10. The other correlations, involving $C$ as determined by Eq. (20), are instead an original depiction of this study and are discussed below.

On the one hand, the negative correlation between $Fr$ and $C/u$ can be justified considering that (i) the Froude number increased during the experiment as a result of the slope increase and (ii) the celerity decreased towards a zero value at morphological equilibrium. On the other hand, one would expect propagation to be enhanced by an increasing velocity, but this may be somehow masked by any scale used for normalization. Indeed, the appearance of $u$ at the numerator of $Fr$ and at the denominator of $C/u$ has determined the obtained trend because with increasing flow velocity, $C$ was increasing less than $u$.

The trends ($\lambda_i/u - C/u$) depict the relationship between the celerity of small perturbations and that of the aggradation wave. The first eigenvalue, $\lambda_1$, is positive and thus directed to the downstream; reasonably, one thus justifies its enhancement of downstream propagation. This eigenvalue is normally attributed to water surface perturbations (De Vries, 1965, 1971, 1973, 1993; Sieben, 1997, 1999; Lyn and Altinakar, 2002; Rosatti and Fraccarollo, 2006; Goutière et al., 2008; Armanini, 2018), therefore a higher celerity of propagation of small perturbations will correspond to a higher celerity of the sediment aggradation wave. The eigenvalues $\lambda_2$ and/or $\lambda_3$ are instead attributed to bed perturbations, even if there is no complete agreement in the literature: as mentioned above, while De Vries (1965, 1971, 1973, 1993) mentioned that just $\lambda_3$ relates to bed surface perturbations, Goutière et al. (2008) and Armanini (2018) argued that both $\lambda_2$ and $\lambda_3$ correspond to bed perturbations, and Lyn and Altinakar (2002) and Sieben (1997,1999) considered that $\lambda_2$ and $\lambda_3$ may be attributed to both bed and water surface perturbations in near-critical conditions. The eigenvalue $\lambda_2$ is directed to the upstream. In numerical studies, the eigenvalues

are associated with the propagation of the effects of the boundary conditions into the domain (e.g., Fasolato et al., 2009). A negative value of $\lambda_2$ would thus indicate that this eigenvalue propagates the downstream boundary condition into the domain. An equilibrium condition is frequently imposed at the downstream end of a modelled channel (i.e., no variation in the bed elevation), which aligns with our experimental setup using a downstream sill (see profiles in Fig. 3(a)). In this context, a higher $|\lambda_2|$ would imply a faster upstream propagation of a condition of minimum elevation, possibly enhancing the downstream propagation of the aggradation wave celerity. Finally, $\lambda_3$ is directed to the downstream, but an increase of $\lambda_3$ determines a reduction of $C$; since $\lambda_3$ is positively correlated with the Froude number, arguments proposed above for the $Fr - C/u$ correlation may also apply to the present one.

In the context of a relationship between the $\lambda$ and $C$, the possibility or not to discard the sediment concentration when expressing the eigenvalues loses its merit. In fact, on the one hand, formulating the eigenvalues discarding the concentration or not changes their values; the value change is even by $50 - 100\%$ for $\lambda_2$, that is the eigenvalue most affected by the sediment concentration. On the other hand, the values of the Pearson correlation coefficient in Table 2 do not change much choosing one option or the other one, and in both cases one can find a relationship between the celerities of the small perturbations and the celerity of the aggradation wave.

The points before the water discharge achieves its nominal value show opposite trends to the others. The approaches introduced above are for unsteady flows, so the different trend cannot be attributed to a limitation of the mathematical depiction. The different trend thus needs to be attributed to the swift change of the flow rate at the very beginning of the experiment.

The above arguments constitute the answers that the present manuscript furnishes to the research questions posed in the Introduction. The results presented for the proof-of-concept experiment are extended to other experiments of the campaign in Supplemental 2.

## 6.3 Process dynamics and technical implications

Morpho-dynamic processes frequently involve a superimposition of different forms. For example, Yalin (1992, p. 12) presented a conceptual sketch with small ripples superimposed to bed-load dunes; Radice (2021) conceptualized a multi-scale propagation for bed-load dunes with different features propagating at different celerity. Furthermore, Zanchi and Radice (2021) mentioned that, in some of their experiments, dunes were superimposed to an aggradation front, even if they did not perform a specific analysis to quantify a different celerity for the dunes. However, as observed in this study, the celerities of the small perturbations and of the aggradation wave are largely different in value, and follow a general trend of 'the smaller, the faster'. From an engineering point of view, the mismatch between the celerity of the small perturbation and that of the aggradation wave implies that one cannot rely on the eigenvalues to determine when sediment supplied into a river will reach any certain point, possibly increasing the hydraulic hazard at that spot. The needed parameter for managing hazard is, instead, the celerity of propagation of the aggradation wave. For the experiment presented in the manuscript, $C/u$ was less than 0.05 (as shown by the scatter plots of Figs. 9 and 11). Also this result is generally confirmed by all the runs of the experimental campaign, as presented in Supplemental 2. Extensive analysis shall reveal how a bulk celerity may depend on the control parameters, following earlier investigations for sediment plumes or fronts under subcritical flow. In this respect, it is noted that the aggradation front celerity measured by Zanchi and Radice (2021) was also generally below 0.05 times the flow velocity characterizing their runs, indicating that this finding may be valid for a wider range of conditions compared to that considered in the present manuscript. Necessary extension of the results of these experimental campaigns to more complicated or field cases requires care, even if the hydro-dynamic conditions applied in this manuscript resemble those of upland rivers with intense sediment transport.

## 7 Conclusions

Riverbed aggradation may quickly occur during intense events, with a significant impact on river morphology and water elevations. In this study, we have explored the aggradation process experimentally, with the aim of characterizing the key temporal scales of propagation. If a high detail is maintained in an aggradation experiment for measuring the bed and water surface elevations in space and time, a corresponding spatial and temporal evolution can be obtained for a dimensionless celerity of propagation of the aggradation wave. This celerity can be, in fact, quantified through the spatial and temporal derivatives of the bed elevation, and by the local and instantaneous flow velocity. Besides, within a quasi-two-phase approach, an experimental quantification can be obtained for the eigenvalues of the governing equations, that represent the celerity of small perturbations. Thanks to this, in the present study we successfully explored the correlation between the celerities of small perturbations and of an aggradation wave, reaching the following major conclusions:

(i) The celerity of aggradation is correlated with the eigenvalues of the governing equations, thus with the celerity of the so-called small perturbations; the general trends show $C$ increasing with $\lambda_1$ and $|\lambda_2|$ and decreasing with $\lambda_3$.

(ii) Even though there is a correlation between the $C$ and $\lambda$, they are largely different in magnitude, with the celerity of the aggradation wave being much smaller than the others.

(iii) From a mathematical point of view, using a model that considers or not the solid concentration may lead to different results for the eigenvalues of the system: while $\lambda_1$ is almost the same for the two approaches and a slight difference exists in $\lambda_3$, a major difference is observed for $\lambda_2$ that has a higher absolute value when the concentration is discarded. However, considering or not the solid concentration in the governing equations does not affect the qualitative relationships between the different celerities, nor the associated correlation coefficients.

(iv) The celerity of aggradation, $C$, is less than 0.05 times the bulk water velocity for the aggradation process that was simulated in the laboratory and presented in this study. This indication, generally confirmed by the other experiments of the campaign, may serve as a preliminary estimation of aggradation wave celerity for engineering purposes.

### Data availability statement

The raw data of the experiment are provided here: https://doi.org/10.5281/zenodo.10641001. Analogous data for all the experiments of the campaign are furnished here: https://doi.org/10.5281/zenodo.14883164.

### Author contribution

Conceptualization: H.E., A.R.; Methodology: H.E., A.R.; Software: E.P., E.Z.; Investigation: all; Formal analysis and Visualization: H.E., E.P., M.H., K.T., M.Y.N., E.Z., R.Z.; Writing – original draft preparation: H.E.; Writing: review & editing: H.E., A.R.; Funding acquisition: A.R.; Project administration: A.R.

### Competing interests

The authors declare that they have no conflict of interest.

### Acknowledgments

The present study has been financially supported by the Italian Ministry of University and Research through the Ph.D. scholarship of H.E. and by the European Union – Next Generation EU through the PRIN project "Sediment Transport REsearch for cAtchments Management" (STREAM), CUP (Univocal Project Code) D53D23003990006. We gratefully acknowledge

Jens Turowski (associate editor), Angel Monsalve (reviewer) and an anonymous reviewer who spent time and effort to stimulate the improvement of the manuscript.

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
