# Peer review of "Investigating the celerity of propagation for small perturbations and dispersive sediment aggradation under a supercritical flow"

_EGUsphere, 2024_

## Referee Comment (RC1)

Based on my review, I recommend that this manuscript be declined in its current form, with an encouragement to resubmit after substantial revision. This recommendation is not based on the scientific merit of the work, which addresses an important topic and presents valuable experimental data, but rather on the need for significant restructuring of the presentation.

The manuscript would benefit from improved organization and flow, as the current structure sometimes creates disconnections between related ideas across paragraphs. Additionally, the separation between results and their interpretation could be clearer, and the discussion section needs substantial development.

The detailed comments below highlight specific instances where improvements are needed, though they should not be considered exhaustive. I encourage the authors to use these comments as guidance for a comprehensive revision. A resubmission that addresses these structural issues would make a valuable contribution to the field

Line 22: Consider revising the term "correlation trends." The figures present scatter plots showing relationships between dimensionless or dimensional variables. To strengthen the analysis, statistical measures (such as correlation coefficients) would be valuable additions to quantify these relationships.

Line 28: Consider strengthening the connection between "This paper is focused..." and the previous discussion of time-scales to improve the logical flow of ideas.

Line 30: Consider reviewing sentence structure throughout the manuscript. The current use of commas occasionally interrupts the natural flow of ideas.

Line 31: The discussion of hazards would be more impactful with specific examples, helping readers better understand the practical implications of this research.

Line 32: Consider expanding the statement "Sediment aggradation has been studied in the past for both the effects of pulsed sediment supply and the formation of depositional fronts" by including key findings from these studies, similar to the effective approach used in Line 39 regarding translational front and dispersive processes.

Line 36: Since celerity is a central concept in this work, consider providing its definition early in the text to establish a clear foundation for readers.

Line 49: The statement "as most (if not all) prior investigations" would be strengthened by citing specific references to support this claim.

Line 51: Consider using consistent terminology throughout the manuscript (e.g., replacing "for example" with "e.g." for consistency).

Line 57: Consider integrating the important explanation "that is the ratio of sediment discharge to the water-sediment mixture discharge" into the main text rather than using parentheses. This would improve readability and emphasize this significant information.

Line 59: Consider introducing the role of sediment concentration with more context, as this is a key parameter that would benefit from a clearer introduction.

Line 60: The use of "However" suggests a contrast - consider clarifying what is being contrasted to strengthen the logical flow.

Line 69: The phrase "something different" could be more specific - consider indicating whether this refers to a different equation, method, or approach.

Line 71: Consider rephrasing "determined as just mentioned" to provide clearer reference to the specific method being discussed.

Line 73: The three research questions presented provide a valuable framework. Consider strengthening how questions 1 and 2 are addressed in the text, as question 3 is well explained but the others would benefit from more explicit answers.

Line 90: Consider integrating parenthetical information into the main text throughout the manuscript. For example, the statement about volumetric concentration could be restructured to maintain better flow while preserving the important reference to Armanini et al. (2009).

Line 104, eq2: A minor technical correction: consider removing the "x" for clarity.

Line 208: The validation statement would be strengthened by providing references and additional details about the validation process.

Line 217: Consider enhancing the experimental description by including key parameters (flow velocity, Froude number, Reynolds number) early in the text. While these are discussed later, providing initial values would help readers better understand the flow regime.

Line 219: Consider focusing figure descriptions on the interpretation of results rather than describing the axes. This would help readers better understand the significance of the findings.

Line 224: The term "relatively high flow velocity" would be more informative with specific values provided.

Line 223: Consider replacing subjective terms like "evident" with more specific descriptions of the observations.

Line 283: Consider replacing subjective phrases like "obviously provides a nicer plot" with more objective descriptions of how the smoothing operation benefits the analysis.

Line 284: The Results section would be strengthened by focusing on specific observations rather than using terms like "evidently." Consider guiding readers through the interpretation with clear, objective descriptions.

Line 283 and 318: Consider incorporating the parenthetical information into the main text, as these details are important for understanding the analysis.

Figure 8: Consider revising the terminology from "correlation" to "scatter plot" to better reflect the analysis presented. To strengthen the relationship analysis, statistical measures (such as Pearson's r or Spearman's rho) could be added. This would allow for quantification of the observed relationships.

Line 324: The reference to "typical curves shown in mathematical studies" would be more helpful with specific examples or references provided.

Figures 9 and 10: Consider revising these figures to:

- Use consistent terminology (scatter plot rather than correlation)
- Maintain consistent axis scales where appropriate for comparison
- Clarify that the data points, rather than graphs, are obtained from the analysis

Line 340: Consider expanding the Discussion section to move beyond restating results, perhaps including broader implications and connections to other studies.

Lines 344 and 353: Consider replacing terms like "obviously" and "evidently" with specific explanations of the observations and their significance.

Line 355: Consider integrating the parenthetical phrase "(positive and thus)" into the main text to improve readability and clarity.

---

## Author Comment (AC1)

**Interactive discussion: Response to comments from Anonymous Referee #1**
November 12th, 2024

**General comment**
This manuscript presents an interesting and valuable contribution to our understanding of sediment transport processes, specifically focusing on the propagation of aggradation waves under supercritical flow conditions. The work addresses an important topic in geomorphology with potential implications for flood risk assessment and channel management. The experimental approach, featuring detailed spatial and temporal measurements of bed elevation changes, provides a strong foundation for investigating the relationship between theoretical predictions of perturbation celerity and observed aggradation patterns.
The study's primary strength lies in its methodical comparison between experimentally observed celerities and theoretical eigenvalues derived from governing equations, offering insights into how different mathematical formulations correlate with physical observations. The laboratory dataset appears robust and well-suited to the investigation's objectives, although only one case was considered.

> Many thanks for the general appreciation of the work.

However, the manuscript would benefit from several improvements to enhance its impact and accessibility. The presentation of the material requires restructuring to improve flow and clarity, particularly in separating results from their interpretation. The discussion section could be expanded to better explore the broader implications of the findings and their potential applications. Additionally, some technical aspects need attention, including more precise definitions of key concepts and a more rigorous approach to statistical analysis.
The discussion section needs to be improved significantly as it appears to repeat the results section without substantially expanding on implications and broader applications, or considering cases beyond the single experiment conducted.
With appropriate revisions, this work has the potential to make a significant contribution to our understanding of sediment transport processes in supercritical flows and provide valuable guidance for practical applications.

> Should the Editor ask us to prepare a revised manuscript, we will follow the Reviewer's suggestions, for which we deeply thank her/him. We just notice here that the experiment presented in this manuscript was actually one among several performed in a campaign that took many months. The key trends found in this study apply also to the other runs of the campaign, even though in this manuscript we preferred to show one as a proof of concept. This will be made explicit in the discussion.

**Introduction to the attached formal review**
Based on my review, I recommend that this manuscript be declined in its current form, with an encouragement to resubmit after substantial revision. This recommendation is not based on the scientific merit of the work, which addresses an important topic and presents valuable experimental data, but rather on the need for significant restructuring of the presentation.
The manuscript would benefit from improved organization and flow, as the current structure sometimes creates disconnections between related ideas across paragraphs. Additionally, the separation between results and their interpretation could be clearer, and the discussion section needs substantial development.

> Honestly, while we do not intend to excuse ourselves from revising the manuscript, we were a bit surprised by a recommendation that the paper be declined, considering the general appreciation of the scientific merit of the work, and that none of the detailed comments below sounds insuperable.

The detailed comments below highlight specific instances where improvements are needed, though they should not be considered exhaustive. I encourage the authors to use these comments as guidance for a comprehensive revision. A resubmission that addresses these structural issues would make a valuable contribution to the field.

**Detailed comments (that we have grouped by topic)**

*Scatter plots and correlation metrics*

Line 22: Consider revising the term "correlation trends." The figures present scatter plots showing relationships between dimensionless or dimensional variables. To strengthen the analysis, statistical measures (such as correlation coefficients) would be valuable additions to quantify these relationships.

Figure 8: Consider revising the terminology from "correlation" to "scatter plot" to better reflect the analysis presented. To strengthen the relationship analysis, statistical measures (such as Pearson's r or Spearman's rho) could be added. This would allow for quantification of the observed relationships.

> In our view, a scatter plot shows the mutual relationship between two quantities, thus how they are correlated. However, we can change from "correlation" to "scatter plot" even if we think that the previous term was also appropriate.
>
> The following table presents the Pearson correlation coefficient to illustrate the strength of the relationship between $C/u$ and $Fr$, as well as $C/u$ and $\lambda i/u$ (Figures 8 and 10). The values of the correlation coefficient for all plots range between 0.5 and 0.6, indicating a moderate correlation between the variables.

| | | *Goutière et al. (2008)* | | | *Morris and Williams (1996)* | | |
|---|---|---|---|---|---|---|---|
| | $Fr$ (Fig. 8) | $\lambda_1/u$ (Fig. 10) | $\lambda_2/u$ (Fig. 10) | $\lambda_3/u$ (Fig. 10) | $\lambda_1/u$ (Fig. 10) | $\lambda_2/u$ (Fig. 10) | $\lambda_3/u$ (Fig. 10) |
| $C/u$ | -0.53 | 0.54 | -0.60 | -0.54 | 0.55 | -0.59 | -0.55 |

*Discussion*

Line 340: Consider expanding the Discussion section to move beyond restating results, perhaps including broader implications and connections to other studies.

> The present discussion goes beyond a repetition of the results as it introduces interpretation, in line with the general comment above that the latter should be separated from the results. Furthermore, it provides the key practical information that C/u is less than 0.04. We will strengthen this as the Reviewer's comments demonstrated that our intent was not easily perceivable by a reader.

*Richer referencing*

Line 31: The discussion of hazards would be more impactful with specific examples, helping readers better understand the practical implications of this research.

> In a revised manuscript, we may add a few lines with examples instead of just referencing papers as in the mentioned line. It is actually now generally agreed that sediment transport is a key factor in flood hazard, and we can indeed support our statements with various examples.

Line 32: Consider expanding the statement "Sediment aggradation has been studied in the past for both the effects of pulsed sediment supply and the formation of depositional fronts" by including key findings from these studies, similar to the effective approach used in Line 39 regarding translational front and dispersive processes.

> We can surely include relevant findings from related studies.
>
> For example, on the topic of pulsed sediment supply, Cui et al. (2003) and Cui and Parker (2005) discussed that in mountain streams sediment enters rivers in pulses, typically from landslides or debris flows. These pulses cause topographic disturbances on the riverbed, which are gradually eliminated through two mechanisms: translation where the topographic high moves downstream, and dispersion where it gradually fades away. By conducting experiments on sediment pulses with varying pulse material and feed sediment sizes, they discovered that in all cases, the pulse deformation was primarily dispersive. Nevertheless, when the pulse was finer than the surrounding sediment, both translation and dispersion were observed. Sklar et al. (2009) conducted a laboratory study to examine the effects of gravel pulses of different volumes and grain sizes on an immobile, armored bed. The results showed that the sediment pulses evolved through a combination of translation and

dispersion, with translation being more pronounced for smaller volumes of added sediment. Finer-grained pulses moved through the flume faster, causing a larger but shorter-lasting bed fining effect.

Furthermore, for the case of more persistent sediment overloading, Soni (1981), Yen et al. (1992), Alves and Cardoso (1999) and Zanchi and Radice (2021) provided quick predictors of a bulk celerity for an aggradation front. These formulae may be used to provide expeditious estimates of the time an aggradation wave would need to move from a sediment source to a key spot.

We will find a suitable equilibrium between adding some details and avoiding excessive length of the added material.

Line 49: The statement "as most (if not all) prior investigations" would be strengthened by citing specific references to support this claim.

We will more sharply mention that the present manuscript indeed considers sediment aggradation in supercritical flow with the purpose of investigating its propagations scales, as, to the best of our knowledge, all prior investigations have been conducted for subcritical conditions.

Line 208: The validation statement would be strengthened by providing references and additional details about the validation process.

Basically what we did was, for some experiments, taking manual readings of bed and water elevation, paying attention to avoiding interference with the field of view of the cameras. The difference between the manual readings and the automatic measurements was satisfactorily small. We will be happy to provide some more information in a revised manuscript.

Line 324: The reference to "typical curves shown in mathematical studies" would be more helpful with specific examples or references provided.

Not sure if citing a specific figure of a referenced paper is appropriate; however, in this statement we referred, for example, to Figure 1 of Lyn and Altinakar (2002) and Figure 2 of Garegnani et al. (2013). Figure 1 of Goutière et al. (2008) could be equally mentioned.

*Contextualization*

Line 28: Consider strengthening the connection between "This paper is focused..." and the previous discussion of time-scales to improve the logical flow of ideas.

This should be also made possible by accounting for a previous comment on line 32.

Line 36: Since celerity is a central concept in this work, consider providing its definition early in the text to establish a clear foundation for readers.

We will strengthen the concept and relevance of celerity, even if we would not take mathematics to the Introduction.

Line 59: Consider introducing the role of sediment concentration with more context, as this is a key parameter that would benefit from a clearer introduction.

This is extensively accounted for in section 2, where approaches for negligible and non-negligible concentration are presented separately. We can spend more words here and, however, mention that more details will be provided in a following section with mathematical models.

Line 73: The three research questions presented provide a valuable framework. Consider strengthening how questions 1 and 2 are addressed in the text, as question 3 is well explained but the others would benefit from more explicit answers.

While we are not answering these questions in the Introduction, answer 1 comes from the mathematical models of section 2 for the small perturbations and from eq. (19) for the aggradation wave. Answer 2 comes from figure 10, and answer 3 from comparing the left and right plots in figures 6 and 10, as well as from figure 9. In a revised manuscript, we would probably rephrase question 1 that needs more specification, and sharpen the answers at the end.

Line 217: Consider enhancing the experimental description by including key parameters (flow velocity, Froude number, Reynolds number) early in the text. While these are discussed later, providing initial values would help readers better understand the flow regime.

We will include the key parameters of the experiment in Table 1 as follows:

| $T$ (s) | $S_0$ (%) | $Q$ (l/s) | $Q_{s-in}$ (m³/s) | $Q_{s0}$ (m³/s) | $Lr$ | $H$ (m) | $U$ (m/s) | $Fr$ | $Re$ |
|---|---|---|---|---|---|---|---|---|---|
| 316 | 1.37 | 7.0 | $4.28\times10^{-4}$ | $1.33\times10^{-4}$ | 3.2 | 0.033 | 0.705 | 1.236 | 19113.2 |

where $H$ and $U$ show water depth and velocity, respectively, calculated using the Gauckler-Strickler formula under the assumption of uniform flow.

*Sharpness*
Line 60: The use of "However" suggests a contrast - consider clarifying what is being contrasted to strengthen the logical flow.
Line 69: The phrase "something different" could be more specific - consider indicating whether this refers to a different equation, method, or approach.
Line 71: Consider rephrasing "determined as just mentioned" to provide clearer reference to the specific method being discussed.
Line 219: Consider focusing figure descriptions on the interpretation of results rather than describing the axes. This would help readers better understand the significance of the findings.
Line 224: The term "relatively high flow velocity" would be more informative with specific values provided.
Line 223: Consider replacing subjective terms like "evident" with more specific descriptions of the observations.
Line 283: Consider replacing subjective phrases like "obviously provides a nicer plot" with more objective descriptions of how the smoothing operation benefits the analysis.
Line 284: The Results section would be strengthened by focusing on specific observations rather than using terms like "evidently." Consider guiding readers through the interpretation with clear, objective descriptions.
Lines 344 and 353: Consider replacing terms like "obviously" and "evidently" with specific explanations of the observations and their significance.
     All these are phrasing modifications that will be implemented to increase rigour and clarity of the manuscript.

*Style issues*
Line 30: Consider reviewing sentence structure throughout the manuscript. The current use of commas occasionally interrupts the natural flow of ideas.
Line 51: Consider using consistent terminology throughout the manuscript (e.g., replacing "for example" with "e.g." for consistency).
Line 57: Consider integrating the important explanation "that is the ratio of sediment discharge to the water-sediment mixture discharge" into the main text rather than using parentheses. This would improve readability and emphasize this significant information.
Line 90: Consider integrating parenthetical information into the main text throughout the manuscript. For example, the statement about volumetric concentration could be restructured to maintain better flow while preserving the important reference to Armanini et al. (2009).
Line 283 and 318: Consider incorporating the parenthetical information into the main text, as these details are important for understanding the analysis.
Line 355: Consider integrating the parenthetical phrase "(positive and thus)" into the main text to improve readability and clarity.
     These are also slight modifications that may improve the flow of ideas.

*Other*
Line 104, eq2: A minor technical correction: consider removing the "x" for clarity.
     Will be done.
Figures 9 and 10: Consider revising these figures to:
- Use consistent terminology (scatter plot rather than correlation)
- Maintain consistent axis scales where appropriate for comparison
- Clarify that the data points, rather than graphs, are obtained from the analysis
We will improve these plots.

---

## Author Comment (AC2)

**Interactive discussion: Response to comments from Anonymous Referee #2**
November 22$^{nd}$, 2024

**General comment**
In the manuscript, Eslami et al. (2024) have investigated the celerity of aggradations in a controlled flume experiment and explored the relationship between the celerity of aggradation waves with the celerity of propagation of small perturbations. This is a nice work and detailed analysis of the experiments; however, I have a number of questions, partly regarding the theoretical conceptualization and clarifications on several points.

> Many thanks for appreciating the work, spending time on it and providing useful comments. We have taken the liberty to group the comments by category (choosing some category titles and changing the comment order), to make our responses more effective.

**Eigenvalues as the celerities of small disturbances**
The authors seem to accept the celerity parameters are eigenvalues of the coefficient matrix of a system of partial differential equations as proposed previously in the literature. I am skeptical about whether these eigenvalues are indeed representing the celerity of a disturbance (of bed or surface water). Thus, I would recommend adding a theoretical background section to introduce how the eigenvalues become celerity from a mathematical perspective. For example, after linearizing the non-linear partial differential equation and establishing a system of differential equations, dU/dt + AdU/dx = 0, explain how this concept translates into the celerity of a disturbance happening at a water surface or streambed. Essentially, you can mention that this problem can be treated as Reimann invariant (e.g., Lyn and Altinakar, 2002) under a characteristic path of dx/dt = lambda, where lambda is characteristic speed or wave speed, which I think is how the term celerity arises. In this path, the disturbance waves move at constant speed over spatial and temporal scales.

> We have indeed presented the system of differential equations in eq. (5) of the manuscript (that is, however, a system of non-linear equations). In a revised manuscript, we will expand the explanation at lines 113-116 by adding the concept of Riemann invariant; thanks for the suggestion.

In the introduction, the concept of celerity of propagation of small perturbations and aggradation wave is not clearly defined. I do not understand what are referred to as small perturbations, and in what criteria the authors define a "small" perturbation here. As I read the introduction of the manuscript, I assumed that the celerity of propagation of small perturbations is in this context, water wave celerity and bed wave celerity. It would be nice to introduce/explain this concept in a clearer way.

> The "small" or "infinitesimal" disturbances are the terms used in the literature when the surface of bed or water are perturbed locally and the resulting waves are of limited amplitude (e.g., Rosatti et al., 2004). As mentioned, the eigenvalues of the coefficient matrix of a system of partial differential equations are interpreted as the celerity of propagation for these local, small-scale waves. In the revised manuscript we will rephrase for clarity.

Eigenvalues are the solutions of characteristic equations that encompass information of matrix coefficients of a system of differential equations. Hence, it is difficult to say, for example, that lambda 1 is celerity of water flow, as this manuscript and previous authors suggested, or lambda 2 and lambda 3 are celerity of bed perturbations. Rather, it may be a combined effect of both water flow and streambed on these eigenvalues. What are the viewpoints of the authors for these parameters, according to the experiment performed here? Additionally, does this characteristic equation det(A-I) = 0 always have 3 real roots? Is there any scenario such that the equation yields two negative roots instead of one, as shown in this paper?

Considering the first question, we have mentioned in the manuscript that there is no complete agreement on how the eigenvalues should be associated to perturbations in the water and bed surfaces (lines 129-132 for De Vries, lines 138-140 for Lyn and Altinakar). In our view, the issue can be hardly solved experimentally because, as mentioned in the previous comment of the Reviewer, the "small perturbations" are not well defined and thus can be hardly observed. We will add this consideration to the revised manuscript.

Since the governing system of equations is hyperbolic, it is well-established that it always produces real and distinct eigenvalues (Rosatti et al., 2004; Rosatti & Fraccarollo, 2006). In agreement with this theory, our experimental results consistently yielded three real eigenvalues. A different behavior may be encountered, instead, with approximate determinations of the eigenvalues. While the approximate solution proposed by Goutière et al. (2008), which is valid for any value of the Froude number, always resulted in three real eigenvalues during our experiments, methods such as those by Lyn and Altinakar (2002) or De Vries (1965, 1971, 1973, 1993) sometimes produced non-real roots. These approximated determinations are not valid across the full range of Froude numbers, which led to inconsistencies in certain conditions.

As per the last question, we have never encountered cases with two negative and one positive eigenvalues.

**Definition of an aggradation celerity and its relationship with the previous one**

As mentioned in the manuscript, the celerity of the aggradation is easily quantified in low Froude number flow since the translational migration of bedload can be captured visually. The authors noted that in the high Froude number flow, sediment and water are moving more dispersive; thus, the celerity of aggradation needs to be more precisely defined before diving into the analysis of the partial differential equations in chapter 2, and later quantified by equation 20.

An aggradation wave represents a larger-scale process compared to "small perturbations" and thus requires a different definition for its celerity of propagation. We will rephrase the Introduction of the revised manuscript to improve clarity.

Line 165, equation 19: I don't understand why setting $dX/dt = 0$ in Equation 19. Does that imply that $C = -(dX/dt)/(dX/dx) = 0$? Later, the authors used bed elevation for their

calculations, but $dz_b/dt$ is not 0, which is a contradiction. Also, in lines 163 and 164, if I understand correctly, celerity is a scalar quantity, while velocity is a vector quantity; they are not the same. On line 67, you mentioned "the celerity of propagation of small perturbations is not the celerity of propagation of the aggradation wave." Then, in equation 19, you defined the local and instantaneous $Cx = dx/dt$, which is the same as the celerity of a disturbance. This is again another contradiction.

The short reply is that there are no contradictions but, evidently, in the manuscript we have not been effective enough in our explanations. We here reply to all the questions of the Reviewer and declare an intent to expand the explanation in the revised manuscript.

Regarding the first question: as mentioned in the manuscript (line 164) the celerity of propagation of a certain quantity $X$ is equal to the velocity at which an observer needs to move to see a constant value for $X$. "Constant value for $X$" then translates into $dX = 0$ and then $dX/dt = 0$. Here we have a total derivation, "d". On the other hand, in eq. (19) we have partial derivatives, "$\partial$". Therefore, not at all a condition that $dX/dt = 0$ will imply that $C = 0$.

We impose $dz_b/dt = 0$ to find the celerity at which any value of the bed elevation migrates, and this is fully appropriate even if the bed elevation varies in both space and time. In this respect, we present the following depiction, that we produced reworking Fig. 2(a) of the manuscript. Here, we see that any $z_b$ value in a profile for a certain time may be found in a profile for larger time, at a downstream location. This indeed demonstrates that a $z_b$ value migrates downstream, at a certain celerity whose determination requires looking for the same value at another space and another time, so exploiting $dz_b/dt = 0$. In the plot, we show values of $z_b$ in a range from 2 cm to 9.5 cm, every 0.5 cm. Migration is depicted by a horizontal displacement from a red circle to a black star (note that a black star may have a red circle around if it becomes the starting point for another displacement).

[Figure]

Figure R1. Migration of $z_b$ values along the channel during the experiment (symbols explained in the text just above).

Third issue: velocity is indeed a vector, but here we are exploring the propagation in 1D conditions. Therefore, the direction of this vector would be confined to the channel's longitudinal direction, effectively reducing it to a scalar in the 1D framework.

Last issue is a contradiction possibly arising due to using $dx/dt$ for every celerity. Actually, indeed a celerity is always a $dx/dt$, since it is a velocity. But different objects may have different celerities (like, for example, a person walking on a

moving train and the train itself). In the manuscript, we wrote (line 383) that "the smaller, the faster". Morphological processes, indeed, are frequently characterized by multiscale propagation. Let us mention a couple of examples from our recent experience. Zanchi and Radice (2021) noticed that, in their aggradation processes under subcritical flow, dunes were sometimes superimposed on depositional forms. The propagation celerity of the dunes may be different from that of the aggradation front. Radice (2021) investigated the propagation of bed-load dunes along a channel at equilibrium, and spotted a multi-celerity propagation considering the dunes, the sediment particles, and the sediment gusts triggered by turbulent flow events. In the present study, an aggradation wave (large object) moves at a different celerity compared to the "small perturbations".

Line 255, equation 20: Is equation 20 derived from equation 19? It seems to me that celerity can be simply defined as C = dx/dt (as in de Vries, 1993; if x is difficult to quantify, you can then use dx/dt = (dX/dt)/(dX/dx) where X is easily to quantify (e.g., bed elevation in the paper). I am not sure about the minus sign in this equation, as it contradicts the standard definition from de Vries or from Morris and Williams.

> We first rewrite the Reviewer's equation with other symbols, because again it is important to see partial derivatives correctly. The Reviewer writes:
>
> $$\frac{\mathrm{d}x}{\mathrm{d}t} = \frac{\partial X/\partial t}{\partial X/\partial x}$$
>
> In a standard case with aggradation, depicted below (fig. R2), this quantity is negative if applied to the bed elevation, because the numerator is positive while the denominator is negative. Since we intend to use these derivatives to depict the downstream propagation of an aggradation wave, the change of sign is appropriate.

[Figure]

Figure R2. Sketch of the aggradation process.

Line 67-68: Why is the celerity of propagation of small perturbations not the celerity of propagation of the aggradation wave? For example, if the aggradation wave is moving downstream and generating disturbances, can these disturbances generate perturbations of water and bed?

> The replies above should have clarified the issue that different objects may move at different velocities. Here we add that the points depicted in Fig. R1 can be used to estimate the celerity of the aggradation wave, since any couple of points (a red circle and a black star) corresponds to a displacement within a certain time and thus to a celerity. The celerity values computed for those couples of points are depicted in Fig. R3 (color bar on the right is for the celerity values). Taking

0.015 m/s as a reference value, it corresponds to C/U = 0.02 for U = 0.705 m/s that is the nominal flow velocity for the experiment. These values are, as expected, in agreement with those of Fig. 5 of the manuscript, obtained from eq. (20); by contrast, they are largely different from those of the eigenvalues (see again Fig. 10 of the manuscript).

[Figure]

Figure R3. Celerities of migrating bed elevations computed from the point couples of Fig. R1.

**Research questions and their answers**

The goal of the present manuscript is not clear according to the results presented there. The authors aim to answer 3 questions. (1) How can one quantify the scales of propagation in an aggradation process? (2) What is the relationship between the aggradation celerity and the celerity of small perturbations? And finally, (3) Which is the impact of considering or not the sediment concentration on the previous point? However, results and discussions were not focused on answering them; instead, the authors presented some graphical results between the aggradation celerity and the celerity of small perturbations, and no specific formula or relationships between have been clearly determined. What are some findings for questions 1 and 2, I did not see it? How one can predict the spatial and temporal scales of an aggradation wave, given the information to solve the system of differential equations here? Having read the title and introduction, I hope the present study can show when and where the aggradations are likely to happen based on the celerity of propagation of aggradations.

Regarding the first question: as mentioned, under high Froude number conditions, the aggradation induced by sediment overloading is dispersive and, as a result, front tracing methods cannot be used to estimate the celerity of the aggradation wave. We thus employed the standard definition of the celerity of a specific quantity, as outlined by Chow et al. (1988) and Jain (2001), which is presented in section 3 of the manuscript. Indeed, by introducing the bed elevation as the specific quantity, we successfully determined the celerity of propagation of the aggradation wave. Figure 5 shows this celerity of propagation in dimensional and dimensionless form. Therefore, the short answer to question 1 is: by applying eq.

(19-20) to the bed elevation. We might rephrase slightly question 1, pointing at supercritical/dispersive aggradation conditions.

Regarding the second question: Figure 10 illustrates the correlation between the dimensionless eigenvalues of the system (representing the celerity of propagation of small perturbations) and the dimensionless celerity of the aggradation wave (determined in section 3). The aim of this work is not to derive a formula explicitly relating the two celerity scales, but rather to demonstrate how these two scales correlate to each other. Based on this, it might be more appropriate to revise the second question to: *"How does the aggradation celerity correlate with the celerity of small perturbations?"*. Furthermore, in response to one of the first reviewer's suggestions, we have computed the Pearson correlation coefficients to quantify the strength of the relationships between $C/u$ and $Fr$, as well as $C/u$ and $\lambda_i/u$ (Figures 8 and 10). The values in the following table range between 0.5 and 0.6, indicating a moderate correlation between the variables. We can add this table and the resulting interpretation to our manuscript to strengthen this aspect of the analysis, and thus the answer to question 2.

| | *Fr* (Fig. 8) | *$\lambda_1/u$* (Fig. 10) | *$\lambda_2/u$* (Fig. 10) | *$\lambda_3/u$* (Fig. 10) | *$\lambda_1/u$* (Fig. 10) | *$\lambda_2/u$* (Fig. 10) | *$\lambda_3/u$* (Fig. 10) |
|---|---|---|---|---|---|---|---|
| | | *Goutière et al. (2008)* | | | *Morris and Williams (1996)* | | |
| *$C/u$* | -0.53 | 0.54 | -0.60 | -0.54 | 0.55 | -0.59 | -0.55 |

Regarding, finally, the third question: in different parts of the manuscript, we have addressed the impact of considering or disregarding sediment concentration on the correlation between the celerity of small perturbations and the celerity of the aggradation wave. For example, in discussion part, line 374 it has been mentioned that: "In the context of a relationship between the $\lambda$ and $C$, the possibility or not to discard the sediment concentration while expressing the eigenvalues loses its merit. Obviously, formulating the eigenvalues with or without concentration changes their values (even by $50 - 100\%$ for $\lambda_2$, that is the eigenvalue most affected by the sediment concentration), but it will be always possible to find a trend linking the celerities of small perturbations and the celerity of the aggradation wave". Similarly, the conclusion chapter briefly summarizes these findings: "From a mathematical point of view, using a model that considers or not the solid concentration may obviously lead to different results for the eigenvalues of the system (while $\lambda_1$ is almost the same for the two approaches and a slight difference exists in $\lambda_3$, a major difference is observed for $\lambda_2$ that has a higher absolute value when the concentration is discarded). However, considering or not solid concentration in the governing equations does not affect the qualitative relationships between the different celerities".

**Interpretation/discussion**

In the discussion section, there is little interpretation of values of lamda1, 2, and 3, as well as c, in the context of the performed experiment. What exactly do these lambdas represent? The authors just presented results with minimal intuition and without

comparison with previous work (i.e., Zanchi and Radice, 2021) to distinguish between subcritical and supercritical cases. I expected to see more on the comparison of how changing lambdas can impact the bed aggradation and bed elevations.

Based on the explanations provided above and in the manuscript, we believe we have addressed the first part of the question regarding the interpretation of eigenvalues and C. We will ensure this is made clearer in the manuscript, particularly by elaborating on the distinction between small-scale waves and large-scale waves, as well as their respective celerities.

Regarding the second part of the comment, it is important to note that the work of Zanchi and Radice (2021) primarily focused on a bulk approach to the issue rather than investigation of the correlation between the two types of celerity of propagation. However, in the reply to a following issue (implications for field conditions) we present the c/U values of the experiments of Zanchi and Radice (2021). It is worth mentioning that the experiment presented in this manuscript is one of several conducted in an extended experimental campaign, the global results of which will be presented in future papers. Basically, in a bulk analysis, we have extracted bulk values of celerity for each experiment and derived a formula predicting the celerity of propagation of the aggradation wave based on the control parameters. However, this is beyond the scope of the present manuscript.

Line 345: I don't quite understand how the correlation trends are obviously consistent. Which one is consistent with the other?

What we intended is the following. Let one consider three quantities, A, B and C. If A increases with B and decreases with C, then B needs to decrease with C. If one finds the latter the trends are consistent; if one instead finds that B increases with C, then there is something wrong. In the present study we found consistent trends; in the revised manuscript we will rephrase to make the statement better understandable.

Line 347: The authors stated: "The second correlation ($Fr$ - $C/u$) has returned the dimensionless celerity as a decreasing function of the Froude number". However, there are some clusters (e.g., 3 lines on top) where C/u increases with respect to Froude number (Figure 8).

As mentioned in line 378, these points correspond to the initial stages of the experiment, where the water discharge had not yet been adjusted to its nominal value. Specifically, the text states: "The points before the water discharge achieves its nominal value show opposite trends to the others. The approaches introduced above are for unsteady flows, so the different trend cannot be attributed to a limitation of the mathematical depiction. The different trend thus needs to be attributed to the swift change of the flow rate at the very beginning of the experiment."

Line 360: If lambda2 and lambda3 are attributed to the bed perturbations, can you explain more on the negative values of lambda 2?

In numerical studies, the eigenvalues are considered to propagate the effect of boundary conditions into the domain (e.g., Fasolato et al., 2009). A negative value

of $\lambda_2$ would thus indicate that this eigenvalue propagates into the domain the boundary condition imposed at the downstream end. An equilibrium condition is frequently imposed at the downstream end of a simulated channel (i.e., no variation in the bed elevation), which is also what we did experimentally with a downstream sill (see profiles in Fig. R1). In this context, a higher $|\lambda_2|$ would tend to transport faster a condition of no aggradation and thus increase the local channel slope and in turn the aggradation celerity. The first version of the manuscript did not contain these arguments, that we could instead add to a revised one.

Line 378: The authors claimed that it is always possible to find a trend linking celerity of small perturbations and celerity of aggradation wave. This is a bold statement that needs to be tested not only in experimental studies but also at field scale, where things are much more complicated. And I am highly skeptical about this.

The statement was related to Fig. 10, where the plots on the left and those on the right have the same shape, even if the eigenvalues may change. So, what we intended to say is that, since considering concentration or not does not change the shape of the point scatter, a correlation would exist in both cases. On the other hand, we do not know if this would hold for any concentration value, thus the Reviewer is right in issuing a warning. In the revised manuscript, we will reduce the boldness of the statement adding the above considerations.

**Reproducibility**
In the result section, I assumed the author performed many experiments and listed the average results or best results, but I only see a table summary of the experiment. Is this experiment (and results) reproducible? Updated: I see now in the results section, the authors mentioned that "This result, shown here for a single experiment, was confirmed by the others run in the current experimental campaign". If the authors run other experiments to confirm the findings here, they should be presented in the manuscript, or at least should be in supplemental information.
Line 215, table 1: It seems that only one experiment was performed (e.g., Table 1). I'm worried about the reproducibility of the results presented in this manuscript.

The key trends found in this study apply also to the other runs of the campaign, even though in this manuscript we preferred to show one as a proof of concept. This will be made explicit in the discussion.

**Implications for field conditions / Engineering relevance**
It is very nice that the authors found the ratio C/u<0.04 based on the experiment. If I interpret it correctly, doesn't this suggest that during high Froude number flows, sediment transport is more dispersive, making aggradation much harder to occur? I wonder what the threshold for the ratio C/u would be under field conditions. If the results presented here are valid, I am curious whether this has implications for understanding aggradation

at field conditions. For example, how long would it take for sediment transport under high Froude number flow to alter riverbed and channel morphology?

The last question of the Reviewer is exactly the initial thrust that motivated an extensive campaign on propagation of aggradation.

Let us tell the full story: Radice et al. (2013) discussed how morphologic processes may impact hydraulic hazard assessment and management in upland environments. They thus performed numerical simulations for a mountain river and found that the aggradation in the downstream portion of the modelled reach (that was an in-town portion, thus a key spot to be considered for flood hazard) was independent on a sediment feeding used as an upstream boundary condition. Then the question was, indeed: how long will it take for the upstream feeding condition to get to the town? Radice and Rosatti (2012), for the same river of the other study, compared the bed profiles obtained for a numerical simulation with a certain upstream sediment yield and for another one with zero yield, and tracked the point at which the two solutions coincided; this point moved at around 50 m/h, which is obviously not a general result and also requires trusting the numerical models, but corresponds to a small percentage of the typical flow velocity in a mountain stream.

Finally, once a suitable laboratory facility was available, experiments have been performed, first in subcritical conditions (Zanchi and Radice, 2021 and other companion works), then in supercritical conditions (the present study and companion works).

By the way, the plots below (Fig. R4) present a manipulation of Zanchi and Radice's data, showing a front celerity (that is actually a bulk celerity values, since in that case a front could be indeed tracked) that, with one exception, was below 0.05 times the flow velocity.

We would not put all this story in the revised manuscript; even if it is the path that took us here, it is probably too long and irrelevant for a reader. Furthermore, we think that the value of 0.04 may be left unchanged in the manuscript to stick to the present experimental conditions, possibly adding a warning that the results should not be generalized at this stage. As mentioned in a previous reply, we are presently working on another manuscript that will aim at being a counterpart of Zanchi and Radice (2021) for supercritical flows, to provide a bulk celerity predictor valid for these conditions.

[Figure]

Figure R4. Dimensionless front celerities for the experiments of Zanchi and Radice (2021).

**Other**

Line 69: What is "something different" here?

> We think that it is actually the definition of eq. (1), then becoming (20). We will rephrase for clarity.

Line 73: Missing parenthesis.

> Will be fixed.

Line 129, equation 10: Lack of definition for Froude number.

> Will be defined.

Line 221: The authors mentioned that they did not set the camera to capture the profile before 140 cm, which piques my curiosity. Was there any aggradation or erosion in this section during the experiment?

> In all the experiments conducted during our campaign, the overloading ratio was consistently greater than 1. As a result, aggradation was observed along the entire length of the channel, thus also for $x < 140$ cm.

Line 320, 335: Were results shown in Figures 8 and 10 for the entire profile of the riverbed in this experiment (e.g., 140 to 520 cm) or only for some selected locations along this section?

> They are for the entire profile of the riverbed (140 to 520 cm).

Line 383: What are other processes in this context?

> We were referring, as examples, to the processes investigated by the referenced studies (long waves for Lanzoni et al., 2006, and bed-load dunes for Radice, 2021). In the revised manuscript, we will revise for clarity.

**References**

Chow, V. T., Maidment, D. R., Mays, L. W. (1988), "Applied hydrology", McGraw-Hill. ISBN: 0070108102.

De Vries M. (1965), "Consideration about non-steady bed-load-transport in open channels", Proc. of the 11th Congress of IAHR, Leningrad, 3.8.1–3.8.8.

De Vries M. (1971), "Solving river problems by hydraulic and mathematical models", Delft Hydraulic Laboratory Publications, Delft, The Netherlands.

De Vries M. (1973), "River-bed variations-aggradation and degradation", Delft Hydraulic Laboratory Publications, Delft, The Netherlands.

De Vries M. (1993), "River Engineering: Lecture notes f10", Delft University of Technology, Faculty of Civil Engineering Department of Hydraulic Engineering, Delft, The Netherlands.

Fasolato G., Ronco P., Di Silvio G. (2009). How fast and how far do variable boundary conditions affect river morphodynamics? Journal of Hydraulic Research, 47(3), 329-339. http://dx.doi.org/10.1080/00221686.2009.9522004

Goutière L., Soares-Frazão S., Savary C., Laraichi T., Zech Y. (2008), "One-Dimensional Model for Transient Flows Involving Bed-Load Sediment Transport and Changes in Flow Regimes", Journal of Hydraulic Engineering, 134(6), 726–735. https://doi.org/10.1061/(ASCE)0733-9429(2008)134:6(726).

Jain, S.C. (2001), "Open-channel flow", John Wiley & Sons, ISBN: 0471356417.

Lanzoni, S., Siviglia, A., Frascati, A., & Seminara, G. (2006), "Long waves in erodible channels and morphodynamic influence", Water Resources Research, 42(6). https://doi:10.1029/2006WR004916.

Lyn, D. A., & Altinakar, M. (2002), "St. Venant–Exner equations for near-critical and transcritical flows", Journal of Hydraulic Engineering, 128(6), 579–587. https://doi.org/10.1061/(ASCE)0733-9429(2002)128:6(579).

Radice A., Rosatti G. (2012), "Sulla modellazione idraulico-morfologica dei corsi d'acqua: il torrente Mallero e la propagazione dell'incertezza legata all'alimentazione solida", *XXXIII Convegno di Idraulica e Costruzioni Idrauliche, Brescia*.

Radice A., Rosatti G., Ballio F., Franzetti S., Mauri M., Spagnolatti M., Garegnani G. (2013), "Management of flood hazard via hydro-morphological river modelling. The case of the Mallero in Italian Alps", *Journal of Flood Risk Management*, Vol. 6, n. 3, 197-209, doi: 10.1111/j.1753-318X.2012.01170.x.

Rosatti G., Fraccarollo L. (2006), "A well-balanced approach for flows over mobile-bed with high sediment-transport", Journal of Computational Physics, 220(1), 312–338, https://doi.org/10.1016/j.jcp.2006.05.012.

Rosatti G., Fraccarollo L., Armanini A. (2004), "Behaviour of small perturbations in 1d mobile-bed models", River Flow, 67–73 (2004).

Zanchi B., Radice A. (2021), "Celerity and height of aggradation fronts in gravel-bed laboratory channel", Journal of Hydraulic Engineering, Vol. 147, n. 10, 04021034, doi: 10.1061/(ASCE)HY.1943-7900.0001923.

---

## Author Response (AR1)

**Investigating the celerity of propagation for small perturbations and dispersive sediment aggradation under a supercritical flow**

Hasan Eslami[1], Erfan Poursoleymanzadeh[1,2], Mojtaba Hiteh[1,3], Keivan Tavakoli[1,4], Melika Yavari Nia[1,5], Ehsan Zadehali[1,6], Reihaneh Zarrabi[1,7], Alessio Radice[1]

[1]Dept. of Civil and Environmental Engineering, Politecnico di Milano, Milan, 20133, Italy
[2]Dept. of Civil Engineering and Management (CEM), University of Twente, Enschede 7522NB, The Netherlands
[3]Dept. Civil and Coastal Engineering, University of Florida, Gainesville, FL 32611, US
[4]Dept of Geography and the Environment, University of Texas at Austin, Austin, TX 78712 – 1697, US
[5]Dept. Soil, Water and Ecosystem Sciences, University of Florida, Gainesville, FL 32611, US
[6]Dept. of Civil, Environmental, and Geo-Engineering, University of Minnesota Twin Cities, Minneapolis, MN 55455, US
[7]Dept. of Geography and Environment, University of Alabama, Tuscaloosa, AL 35487, US

**Response to the Reviewers' comments**

We thank the Reviewers for their appreciation of the manuscript content and for the comments they provided. We are now resubmitting the manuscript, with the following major modifications:
- The introduction has been extensively revised, particularly (1) describing in more detail some prior literature studies, (2) improving the presentation of temporal scales and celerity for the small perturbations and aggradation wave, (3) slightly rephrasing the research questions.
- We have revised the presentation of the results and the discussion. In the latter, we have added some further comparison with other works and comments on the engineering relevance of the present findings.
- Even though we have finally resolved to maintain a single experiment in the manuscript as a proof of concept, a supplemental file 2 has been added with results for the other experiments of the campaign, to prove that the findings presented in the paper are valid also for the other runs. The supplemental file is mentioned from the very beginning (in the abstract), so that a reader will not miss it.

All the comments of the Reviewers have been accounted for, as described in this document. We are confident that this improved version of the manuscript will be appreciated by the reviewers and editors. We submit a track-change version of the manuscript, and a version with changes accepted for easier reading. Line numbers in this letter refer to the version with the modifications accepted.

**Response to comments from Anonymous Referee #1**

**General comment**
This manuscript presents an interesting and valuable contribution to our understanding of sediment transport processes, specifically focusing on the propagation of aggradation waves under supercritical flow conditions. The work addresses an important topic in geomorphology with potential implications for flood risk assessment and channel management. The experimental approach, featuring detailed spatial and temporal measurements of bed elevation changes, provides a strong foundation for investigating the relationship between theoretical predictions of perturbation celerity and observed aggradation patterns.
The study's primary strength lies in its methodical comparison between experimentally observed celerities and theoretical eigenvalues derived from governing equations, offering insights into how different mathematical formulations correlate with physical observations. The laboratory dataset appears robust and well-suited to the investigation's objectives, although only one case was considered.

> Many thanks for the general appreciation of the work.

However, the manuscript would benefit from several improvements to enhance its impact and accessibility. The presentation of the material requires restructuring to improve flow and clarity, particularly in separating results from their interpretation. The discussion section could be expanded to better explore the broader implications of the findings and their potential applications. Additionally, some technical aspects need attention, including more precise definitions of key concepts and a more rigorous approach to statistical analysis.
The discussion section needs to be improved significantly as it appears to repeat the results section without substantially expanding on implications and broader applications, or considering cases beyond the single experiment conducted.
With appropriate revisions, this work has the potential to make a significant contribution to our understanding of sediment transport processes in supercritical flows and provide valuable guidance for practical applications.

> The comments of the Reviewer have been accounted for and incorporated into the revised manuscript, as described in the detailed responses below.

**Introduction to the attached formal review**
Based on my review, I recommend that this manuscript be declined in its current form, with an encouragement to resubmit after substantial revision. This recommendation is not based on the scientific merit of the work, which addresses an important topic and presents valuable experimental data, but rather on the need for significant restructuring of the presentation.
The manuscript would benefit from improved organization and flow, as the current structure sometimes creates disconnections between related ideas across paragraphs. Additionally, the separation between results and their interpretation could be clearer, and the discussion section needs substantial development.

> Honestly, while we did not intend to excuse ourselves from revising the manuscript (as demonstrated by the amount of revisions applied to the manuscript), we were a bit surprised by a recommendation that the paper be declined, considering the general appreciation of the scientific merit of the work. We trust that the Reviewer will appreciate the effort we have put into the revision.

The detailed comments below highlight specific instances where improvements are needed, though they should not be considered exhaustive. I encourage the authors to use these comments as guidance for a comprehensive revision. A resubmission that addresses these structural issues would make a valuable contribution to the field.

> See below for detailed responses.

**Detailed comments (that we have grouped by topic)**
*Scatter plots and correlation metrics*

Line 22: Consider revising the term "correlation trends." The figures present scatter plots showing relationships between dimensionless or dimensional variables. To strengthen the analysis, statistical measures (such as correlation coefficients) would be valuable additions to quantify these relationships.

Figure 8: Consider revising the terminology from "correlation" to "scatter plot" to better reflect the analysis presented. To strengthen the relationship analysis, statistical measures (such as Pearson's r or Spearman's rho) could be added. This would allow for quantification of the observed relationships.

> We have revised the text and used "scatter plot" at several instances (e.g., lines 25, 366, 373, 380, 383, 388, 406).
>
> We have added a table (Table 2) with the values of the Pearson correlation coefficient, whose values are commented at lines 394-396, 409, 439. Furthermore, as mentioned in the text, the newly added Supplemental 2 contains analogous assessments for other experiments of the campaign.

*Discussion*

Line 340: Consider expanding the Discussion section to move beyond restating results, perhaps including broader implications and connections to other studies.

> First, in the revised manuscript we have clarified that in the discussion our intent was indeed to go beyond a repetition of results, that are interpreted (lines 406-441).
>
> Second, the outlook has been expanded and is provided at lines 448-464.

*Richer referencing*

Line 31: The discussion of hazards would be more impactful with specific examples, helping readers better understand the practical implications of this research.

> We have provided a more detailed description of a few literature studies at lines 39-43.

Line 32: Consider expanding the statement "Sediment aggradation has been studied in the past for both the effects of pulsed sediment supply and the formation of depositional fronts" by including key findings from these studies, similar to the effective approach used in Line 39 regarding translational front and dispersive processes.

> Also for this suggestion, some lines have been added (51-58).

Line 49: The statement "as most (if not all) prior investigations" would be strengthened by citing specific references to support this claim.

> We have rephrased the statement (lines 67-69).

Line 208: The validation statement would be strengthened by providing references and additional details about the validation process.

> Information has been added (lines 246-253).

Line 324: The reference to "typical curves shown in mathematical studies" would be more helpful with specific examples or references provided.

> At line 376, we now refer to specific figures in the referenced studies.

*Contextualization*

Line 28: Consider strengthening the connection between "This paper is focused..." and the previous discussion of time-scales to improve the logical flow of ideas.

> In order to account for this comment, rephrasing has been applied at lines 35-39 to smoothen the flow of ideas.

Line 36: Since celerity is a central concept in this work, consider providing its definition early in the text to establish a clear foundation for readers.

> After deep thinking, we have resolved to avoid mathematics in the introduction. However, the qualitative definition and technical relevance of celerity are now more evident, thanks to the revision at lines 43-48.

Line 59: Consider introducing the role of sediment concentration with more context, as this is a key parameter that would benefit from a clearer introduction.

> Also for this comment, we have preferred to keep mathematics out of the introduction. However, some revision has been made at lines 81-86.

Line 73: The three research questions presented provide a valuable framework. Consider strengthening how questions 1 and 2 are addressed in the text, as question 3 is well explained but the others would benefit from more explicit answers.

> We have rephrased a bit questions 1 and 2 (lines 98-100). Furthermore, at other places of the manuscript, we establish a connection between the findings and the initial questions (lines 308-310, 388-393).

Line 217: Consider enhancing the experimental description by including key parameters (flow velocity, Froude number, Reynolds number) early in the text. While these are discussed later, providing initial values would help readers better understand the flow regime.

> We have included the key parameters of the experiment in Table 1.

*Sharpness*

Line 60: The use of "However" suggests a contrast - consider clarifying what is being contrasted to strengthen the logical flow.

> Revised (line 86).

Line 69: The phrase "something different" could be more specific - consider indicating whether this refers to a different equation, method, or approach.

> Revised (lines 96-97).

Line 71: Consider rephrasing "determined as just mentioned" to provide clearer reference to the specific method being discussed.

> This sentence has been removed.

Line 219: Consider focusing figure descriptions on the interpretation of results rather than describing the axes. This would help readers better understand the significance of the findings.

> Revision has been made (lines 267-274).

Line 224: The term "relatively high flow velocity" would be more informative with specific values provided.

> A mention to the velocity value reported in Table 1 has been added at line 274.

Line 223: Consider replacing subjective terms like "evident" with more specific descriptions of the observations.

Line 283: Consider replacing subjective phrases like "obviously provides a nicer plot" with more objective descriptions of how the smoothing operation benefits the analysis.

Line 284: The Results section would be strengthened by focusing on specific observations rather than using terms like "evidently." Consider guiding readers through the interpretation with clear, objective descriptions.

Lines 344 and 353: Consider replacing terms like "obviously" and "evidently" with specific explanations of the observations and their significance.

> At multiple places, we have removed these examples of non-scientific wording.

*Style issues*

Line 30: Consider reviewing sentence structure throughout the manuscript. The current use of commas occasionally interrupts the natural flow of ideas.

Line 51: Consider using consistent terminology throughout the manuscript (e.g., replacing "for example" with "e.g." for consistency).

Line 57: Consider integrating the important explanation "that is the ratio of sediment discharge to the water-sediment mixture discharge" into the main text rather than using parentheses. This would improve readability and emphasize this significant information.

Line 90: Consider integrating parenthetical information into the main text throughout the manuscript. For example, the statement about volumetric concentration could be restructured to maintain better flow while preserving the important reference to Armanini et al. (2009).

Line 283 and 318: Consider incorporating the parenthetical information into the main text, as these details are important for understanding the analysis.

Line 355: Consider integrating the parenthetical phrase "(positive and thus)" into the main text to improve readability and clarity.

> We have revised and avoided putting too much text within () throughout the manuscript.

*Other*

Line 104, eq2: A minor technical correction: consider removing the "x" for clarity.

Done.

Figures 9 and 10: Consider revising these figures to:
- Use consistent terminology (scatter plot rather than correlation)
- Maintain consistent axis scales where appropriate for comparison
- Clarify that the data points, rather than graphs, are obtained from the analysis
The plots have been improved.

**Response to comments from Anonymous Referee #2**

**General comment**
In the manuscript, Eslami et al. (2024) have investigated the celerity of aggradations in a controlled flume experiment and explored the relationship between the celerity of aggradation waves with the celerity of propagation of small perturbations. This is a nice work and detailed analysis of the experiments; however, I have a number of questions, partly regarding the theoretical conceptualization and clarifications on several points.

> Many thanks for appreciating the work, spending time on it and providing useful comments. We have taken the liberty to group the comments by category (choosing some category titles and changing the comment order), to make our responses more effective.

**Eigenvalues as the celerities of small disturbances**
The authors seem to accept the celerity parameters are eigenvalues of the coefficient matrix of a system of partial differential equations as proposed previously in the literature. I am skeptical about whether these eigenvalues are indeed representing the celerity of a disturbance (of bed or surface water). Thus, I would recommend adding a theoretical background section to introduce how the eigenvalues become celerity from a mathematical perspective. For example, after linearizing the non-linear partial differential equation and establishing a system of differential equations, dU/dt + AdU/dx = 0, explain how this concept translates into the celerity of a disturbance happening at a water surface or streambed. Essentially, you can mention that this problem can be treated as Reimann invariant (e.g., Lyn and Altinakar, 2002) under a characteristic path of dx/dt = lambda, where lambda is characteristic speed or wave speed, which I think is how the term celerity arises. In this path, the disturbance waves move at constant speed over spatial and temporal scales.

> Section 2 of the manuscript indeed contains the mathematical ground (maybe this comment of the Reviewer was written while reading the introduction). The interpretation of the eingevalues as celerities appears at lines 140-146, where we have also added a mention to the Reimann invariant problem.

In the introduction, the concept of celerity of propagation of small perturbations and aggradation wave is not clearly defined. I do not understand what are referred to as small perturbations, and in what criteria the authors define a "small" perturbation here. As I read the introduction of the manuscript, I assumed that the celerity of propagation of small perturbations is in this context, water wave celerity and bed wave celerity. It would be nice to introduce/explain this concept in a clearer way.

> Following the suggestion, we have expanded the description/definition of the two types of celerity. The "small" or "infinitesimal" disturbances are now introduced at lines 78-81, while the aggradation wave as a large-scale disturbance comes at lines 92-97.

Eigenvalues are the solutions of characteristic equations that encompass information of matrix coefficients of a system of differential equations. Hence, it is difficult to say, for

example, that lambda 1 is celerity of water flow, as this manuscript and previous authors suggested, or lambda 2 and lambda 3 are celerity of bed perturbations. Rather, it may be a combined effect of both water flow and streambed on these eigenvalues. What are the viewpoints of the authors for these parameters, according to the experiment performed here? Additionally, does this characteristic equation det(A-I) = 0 always have 3 real roots? Is there any scenario such that the equation yields two negative roots instead of one, as shown in this paper?

> Considering the first question, we have mentioned in the manuscript that there is no complete agreement on how the eigenvalues should be associated to perturbations in the water and bed surfaces (see lines 163-166 for De Vries, 173-174 for Lyn and Altinakar, 420-427 for a summary within the discussion section).
>
> Since the governing system of equations is hyperbolic, it is well-established that it always produces real and distinct eigenvalues. In agreement with this theory, our experimental results consistently yielded three real eigenvalues. A different behavior may be encountered, instead, with approximate determinations of the eigenvalues. While the approximate solution proposed by Goutière et al. (2008), which is valid for any value of the Froude number, always resulted in three real eigenvalues during our experiments, methods such as those by Lyn and Altinakar (2002) or De Vries (1965, 1971, 1973, 1993) sometimes produced non-real roots. These approximated determinations are not valid across the full range of Froude numbers, which led to inconsistencies in certain conditions. These issues are briefly mentioned at lines 158 and 314.
>
> As per the last question, we have never encountered cases with two negative and one positive eigenvalues. Such a situation is never mentioned in the literature; thus, no modification has been made to the manuscript in this respect.

**Definition of an aggradation celerity and its relationship with the previous one**

As mentioned in the manuscript, the celerity of the aggradation is easily quantified in low Froude number flow since the translational migration of bedload can be captured visually. The authors noted that in the high Froude number flow, sediment and water are moving more dispersive; thus, the celerity of aggradation needs to be more precisely defined before diving into the analysis of the partial differential equations in chapter 2, and later quantified by equation 20.

> An aggradation wave represents a larger-scale process compared to "small perturbations" and thus requires a different definition for its celerity of propagation. Lines 92-97 of the revised manuscript provide this concept.

Line 165, equation 19: I don't understand why setting dX/dt = 0 in Equation 19. Does that imply that C = -(dX/dt)/(dX/dx) = 0? Later, the authors used bed elevation for their calculations, but $dz_b/dt$ is not 0, which is a contradiction. Also, in lines 163 and 164, if I understand correctly, celerity is a scalar quantity, while velocity is a vector quantity; they are not the same. On line 67, you mentioned "the celerity of propagation of small perturbations is not the celerity of propagation of the aggradation wave." Then, in equation

19, you defined the local and instantaneous Cx = dx/dt, which is the same as the celerity of a disturbance. This is again another contradiction.

The short reply is that there are no contradictions but, evidently, in the manuscript we have not been effective enough in our explanations. Some changes have been made to make things clearer.

Regarding the first question: as mentioned in the manuscript (lines 196-198) the celerity of propagation of a certain quantity $X$ is equal to the velocity at which an observer needs to move to see a constant value for $X$. "Constant value for $X$" then translates into $dX = 0$ and then $dX/dt = 0$. Here we have a total derivation, "d". On the other hand, in eq. (19) we have partial derivatives, "$\partial$". Therefore, not at all a condition that $dX/dt = 0$ will imply that C = 0. In the revised manuscript, we have expanded the arguments at lines 198-200. At line 196, we remark that this derivation is borrowed from highly knowledgeable books.

We impose $dz_b/dt = 0$ to find the celerity at which any value of the bed elevation migrates, and this is fully appropriate even if the bed elevation varies in both space and time. In this respect, we present the following depiction, that we produced reworking Fig. 2(a) of the manuscript. Here, we see that any $z_b$ value in a profile for a certain time may be found in a profile for larger time, at a downstream location. This indeed demonstrates that a $z_b$ value migrates downstream, at a certain celerity whose determination requires looking for the same value at another space and another time, so exploiting $dz_b/dt = 0$. In the plot, we show values of $z_b$ in a range from 2 cm to 9.5 cm, every 0.5 cm. Migration is depicted by a horizontal displacement from a red circle to a black star (note that a black star may have a red circle around if it becomes the starting point for another displacement). In the revised manuscript, we deal with this issue at lines 203-204 and 301-303.

[Figure]

Figure R1. Migration of $z_b$ values along the channel during the experiment (symbols explained in the text just above).

Third issue: velocity is indeed a vector, but here we are exploring the propagation in 1D conditions. Therefore, the direction of this vector would be confined to the channel's longitudinal direction, effectively reducing it to a scalar in the 1D framework. No change has been made to the manuscript in this respect, as the framework is declared to be one-dimensional from the beginning of section 2 (line 110) and, consistently, also the u velocity in eq. (1) is a scalar.

Last issue is a contradiction possibly arising due to using $dx/dt$ for every celerity. Actually, indeed a celerity is always a $dx/dt$, since it is a velocity. But different objects may have different celerities (like, for example, a person walking on a

moving train and the train itself). In the manuscript, we wrote that "the smaller, the faster". Morphological processes, indeed, are frequently characterized by multiscale propagation. Let us mention a couple of examples from our recent experience. Zanchi and Radice (2021) noticed that, in their aggradation processes under subcritical flow, dunes were sometimes superimposed on depositional forms. The propagation celerity of the dunes may be different from that of the aggradation front. Radice (2021) investigated the propagation of bed-load dunes along a channel at equilibrium, and spotted a multi-celerity propagation considering the dunes, the sediment particles, and the sediment gusts triggered by turbulent flow events. In the present study, an aggradation wave (large object) moves at a different celerity compared to the "small perturbations". In the revised manuscript, we provide a brief discussion of this issue at lines 448-453.

Line 255, equation 20: Is equation 20 derived from equation 19? It seems to me that celerity can be simply defined as C = dx/dt (as in de Vries, 1993; if x is difficult to quantify, you can then use dx/dt = (dX/dt)/(dX/dx) where X is easily to quantify (e.g., bed elevation in the paper). I am not sure about the minus sign in this equation, as it contradicts the standard definition from de Vries or from Morris and Williams.

> We first rewrite the Reviewer's equation with other symbols, because again it is important to see partial derivatives correctly. The Reviewer writes:
>
> $$\frac{\mathrm{d}x}{\mathrm{d}t} = \frac{\partial X/\partial t}{\partial X/\partial x}$$
>
> In a standard case with aggradation, depicted below (fig. R2), this quantity is negative if applied to the bed elevation, because the numerator is positive while the denominator is negative. Since we intend to use these derivatives to depict the downstream propagation of an aggradation wave, the change of sign is appropriate. The changes made to the revised manuscript and described above (lines 198-200, 306-310) should be already enough to account for this comment.

[Figure]

Figure R2. Sketch of the aggradation process.

Line 67-68: Why is the celerity of propagation of small perturbations not the celerity of propagation of the aggradation wave? For example, if the aggradation wave is moving downstream and generating disturbances, can these disturbances generate perturbations of water and bed?

> The replies above should have clarified the issue that different objects may move at different velocities (particularly the discussion at lines 448-453). Here we add that the points depicted in Fig. R1 can be used to estimate the celerity of the aggradation wave, since any couple of points (a red circle and a black star)

corresponds to a displacement within a certain time and thus to a celerity. The celerity values computed for those couples of points are depicted in Fig. R3 (color bar on the right is for the celerity values). Taking 0.015 m/s as a reference value, it corresponds to C/U = 0.02 for U = 0.705 m/s that is the nominal flow velocity for the experiment. These values are, as expected, in agreement with those of Fig. 5 of the manuscript, obtained from eq. (20); by contrast, they are largely different from those of the eigenvalues (see again Fig. 10 of the manuscript). This second part of the exercise is mentioned at lines 306-310 to further support the validity of eq. (20).

[Figure]

Figure R3. Celerities of migrating bed elevations computed from the point couples of Fig. R1.

**Research questions and their answers**

The goal of the present manuscript is not clear according to the results presented there. The authors aim to answer 3 questions. (1) How can one quantify the scales of propagation in an aggradation process? (2) What is the relationship between the aggradation celerity and the celerity of small perturbations? And finally, (3) Which is the impact of considering or not the sediment concentration on the previous point? However, results and discussions were not focused on answering them; instead, the authors presented some graphical results between the aggradation celerity and the celerity of small perturbations, and no specific formula or relationships between have been clearly determined. What are some findings for questions 1 and 2, I did not see it? How one can predict the spatial and temporal scales of an aggradation wave, given the information to solve the system of differential equations here? Having read the title and introduction, I hope the present study can show when and where the aggradations are likely to happen based on the celerity of propagation of aggradations.

We have rephrased a bit questions 1 and 2 (lines 98-100). Furthermore, at other places of the manuscript, we establish a connection between the findings and the initial questions (lines 308-310, 388-393).

Regarding the first question: as mentioned, under high Froude number conditions, the aggradation induced by sediment overloading is dispersive and, as a result,

front tracing methods cannot be used to estimate the celerity of the aggradation wave. We thus employed the standard definition of the celerity of a specific quantity, as outlined by Chow et al. (1988) and Jain (2001), which is presented in section 3 of the manuscript. Indeed, by introducing the bed elevation as the specific quantity, we successfully determined the celerity of propagation of the aggradation wave. Figure 5 shows this celerity of propagation in dimensional and dimensionless form. Therefore, the short answer to question 1 is: by applying eq. (19-20) to the bed elevation.

Regarding the second question: Figure 10 illustrates the correlation between the dimensionless eigenvalues of the system (representing the celerity of propagation of small perturbations) and the dimensionless celerity of the aggradation wave (determined in section 3). The aim of this work is not to derive a formula explicitly relating the two celerity scales, but rather to demonstrate how these two scales correlate to each other. Furthermore, in response to one of the first reviewer's suggestions, we have computed the Pearson correlation coefficients to quantify the strength of the relationships between $C/u$ and $Fr$, as well as $C/u$ and $\lambda_i/u$ (Figures 8 and 10). The values of the correlation coefficient come with Table 2 of the revised manuscript and, for other experiments, in the newly added Supplemental 2.

Regarding, finally, the third question: in different parts of the manuscript, we have addressed the impact of considering or disregarding sediment concentration on the correlation between the celerity of small perturbations and the celerity of the aggradation wave (see the lines mentioned at the beginning of this reply).

**Interpretation/discussion**

In the discussion section, there is little interpretation of values of lamda1, 2, and 3, as well as c, in the context of the performed experiment. What exactly do these lambdas represent? The authors just presented results with minimal intuition and without comparison with previous work (i.e., Zanchi and Radice, 2021) to distinguish between subcritical and supercritical cases. I expected to see more on the comparison of how changing lambdas can impact the bed aggradation and bed elevations.

Based on the explanations provided above and in the manuscript, we believe we have addressed the first part of the question regarding the interpretation of eigenvalues and C. In the revised manuscript, the interpretation comes at lines 406-441 of the discussion.

Regarding the second part of the comment, it is important to note that the work of Zanchi and Radice (2021) primarily focused on a bulk approach to the issue rather than investigation of the correlation between the two types of celerity of propagation. However, in the reply to a following issue (implications for field conditions) we present the c/U values of the experiments of Zanchi and Radice (2021) and describe the changes made to the revised manuscript. It is worth mentioning that the experiment presented in this manuscript is one of several conducted in an extended experimental campaign, the global results of which will be presented in future papers. Basically, in a wide analysis, we will determine bulk

values of celerity for each experiment and derive a formula predicting the celerity of propagation of the aggradation wave based on the control parameters. However, this is beyond the scope of the present manuscript.

Line 345: I don't quite understand how the correlation trends are obviously consistent. Which one is consistent with the other?
In the revised manuscript, this has been explained at lines 407-409.

Line 347: The authors stated: "The second correlation ($Fr$ - $C/u$) has returned the dimensionless celerity as a decreasing function of the Froude number". However, there are some clusters (e.g., 3 lines on top) where C/u increases with respect to Froude number (Figure 8).
These points correspond to the initial stages of the experiment, where the water discharge had not yet been adjusted to its nominal value. The issue is clarified at lines 370-371 and 442-444.

Line 360: If lambda2 and lambda3 are attributed to the bed perturbations, can you explain more on the negative values of lambda 2?
In numerical studies, the eigenvalues are considered to propagate the effect of boundary conditions into the domain (e.g., Fasolato et al., 2009). A negative value of $\lambda_2$ would thus indicate that this eigenvalue propagates into the domain the boundary condition imposed at the downstream end. An equilibrium condition is frequently imposed at the downstream end of a simulated channel (i.e., no variation in the bed elevation), which is also what we did experimentally with a downstream sill (see profiles in Fig. R1). In this context, a higher $|\lambda_2|$ would tend to transport faster a condition of no aggradation and thus increase the local channel slope and in turn the aggradation celerity. In the revised manuscript, we have added this material at lines 427-433.

Line 378: The authors claimed that it is always possible to find a trend linking celerity of small perturbations and celerity of aggradation wave. This is a bold statement that needs to be tested not only in experimental studies but also at field scale, where things are much more complicated. And I am highly skeptical about this.
The statement was related to Fig. 10, where the plots on the left and those on the right have the same shape, even if the eigenvalues may change. So, what we intended to say is that, since considering concentration or not does not change the shape of the point scatter, a correlation would exist in both cases. However, in the revised manuscript we have modified the statement by revising lines 436-441. Furthermore, the addition of Supplemental 2 strengthens the findings, at least for our experiments. A mention to the transfer to field scale has been added at lines 462-464.

**Reproducibility**

In the result section, I assumed the author performed many experiments and listed the average results or best results, but I only see a table summary of the experiment. Is this experiment (and results) reproducible? Updated: I see now in the results section, the authors mentioned that "This result, shown here for a single experiment, was confirmed by the others run in the current experimental campaign". If the authors run other experiments to confirm the findings here, they should be presented in the manuscript, or at least should be in supplemental information.
Line 215, table 1: It seems that only one experiment was performed (e.g., Table 1). I'm worried about the reproducibility of the results presented in this manuscript.

> The key trends found in this study apply also to the other runs of the campaign, even though in this manuscript we preferred to show one as a proof of concept; this is explicitly mentioned at lines 207-208 and 446-447. However, at these same lines we mention that we have also produced a supplementary file (Supplemental 2), in this way fully following the Reviewer's suggestion.

**Implications for field conditions / Engineering relevance**
It is very nice that the authors found the ratio C/u<0.04 based on the experiment. If I interpret it correctly, doesn't this suggest that during high Froude number flows, sediment transport is more dispersive, making aggradation much harder to occur? I wonder what the threshold for the ratio C/u would be under field conditions. If the results presented here are valid, I am curious whether this has implications for understanding aggradation at field conditions. For example, how long would it take for sediment transport under high Froude number flow to alter riverbed and channel morphology?

> The last question of the Reviewer is exactly the initial thrust that motivated an extensive campaign on propagation of aggradation.
> Let us tell the our story: Radice et al. (2013) discussed how morphologic processes may impact hydraulic hazard assessment and management in upland environments. They thus performed numerical simulations for a mountain river and found that the aggradation in the downstream portion of the modelled reach (that was an in-town portion, thus a key spot to be considered for flood hazard) was independent on a sediment feeding used as an upstream boundary condition. Then the question was, indeed: how long will it take for the upstream feeding condition to get to the town? Radice and Rosatti (2012), for the same river of the other study, compared the bed profiles obtained for a numerical simulation with a certain upstream sediment yield and for another one with zero yield, and tracked the point at which the two solutions coincided; this point moved at around 50 m/h, which is obviously not a general result and also requires trusting the numerical models, but corresponds to a small percentage of the typical flow velocity in a mountain stream.
> Finally, once a suitable laboratory facility was available, experiments have been performed, first in subcritical conditions (Zanchi and Radice, 2021 and other companion works), then in supercritical conditions (the present study and companion works).

By the way, the plots below (Fig. R4) present a manipulation of Zanchi and Radice's (2021) data, showing a front celerity (that is actually a bulk celerity values, since in that case a front could be indeed tracked) that, with one exception, was below 0.05 times the flow velocity.

We have not put all this story in the revised manuscript; even if it is the path that took us here, it is probably too long and irrelevant for a reader. Still, in the revised manuscript, we have expanded the engineering relevance of the aggradation wave celerity (lines 35-48) and we have added some outlook material (lines 454-464).

[Figure]

Figure R4. Dimensionless front celerities for the experiments of Zanchi and Radice (2021).

**Other**

Line 69: What is "something different" here?
    Revised (lines 96-97).

Line 73: Missing parenthesis.
    The statement has been removed.

Line 129, equation 10: Lack of definition for Froude number.
    Definition provided at line 158.

Line 221: The authors mentioned that they did not set the camera to capture the profile before 140 cm, which piques my curiosity. Was there any aggradation or erosion in this section during the experiment?
    Aggradation happened also upstream of the focus reach. Further description has been added at lines 267-271.

Line 320, 335: Were results shown in Figures 8 and 10 for the entire profile of the riverbed in this experiment (e.g., 140 to 520 cm) or only for some selected locations along this section?
    They are for the entire profile of the riverbed (140 to 520 cm). Information has been added at line 369.

Line 383: What are other processes in this context?
    This text has been rephrased (lines 448-453).

**References**

Chow, V. T., Maidment, D. R., Mays, L. W. (1988), "Applied hydrology", McGraw-Hill. ISBN: 0070108102.

De Vries M. (1965), "Consideration about non-steady bed-load-transport in open channels", Proc. of the 11th Congress of IAHR, Leningrad, 3.8.1–3.8.8.

De Vries M. (1971), "Solving river problems by hydraulic and mathematical models", Delft Hydraulic Laboratory Publications, Delft, The Netherlands.

De Vries M. (1973), "River-bed variations-aggradation and degradation", Delft Hydraulic Laboratory Publications, Delft, The Netherlands.

De Vries M. (1993), "River Engineering: Lecture notes f10", Delft University of Technology, Faculty of Civil Engineering Department of Hydraulic Engineering, Delft, The Netherlands.

Fasolato G., Ronco P., Di Silvio G. (2009). How fast and how far do variable boundary conditions affect river morphodynamics? Journal of Hydraulic Research, 47(3), 329-339. http://dx.doi.org/10.1080/00221686.2009.9522004

Goutière L., Soares-Frazão S., Savary C., Laraichi T., Zech Y. (2008), "One-Dimensional Model for Transient Flows Involving Bed-Load Sediment Transport and Changes in Flow Regimes", Journal of Hydraulic Engineering, 134(6), 726–735. https://doi.org/10.1061/(ASCE)0733-9429(2008)134:6(726).

Jain, S.C. (2001), "Open-channel flow", John Wiley & Sons, ISBN: 0471356417.

Lanzoni, S., Siviglia, A., Frascati, A., & Seminara, G. (2006), "Long waves in erodible channels and morphodynamic influence", Water Resources Research, 42(6). https://doi:10.1029/2006WR004916.

Lyn, D. A., & Altinakar, M. (2002), "St. Venant–Exner equations for near-critical and transcritical flows", Journal of Hydraulic Engineering, 128(6), 579–587. https://doi.org/10.1061/(ASCE)0733-9429(2002)128:6(579).

Radice A., Rosatti G. (2012), "Sulla modellazione idraulico-morfologica dei corsi d'acqua: il torrente Mallero e la propagazione dell'incertezza legata all'alimentazione solida", *XXXIII Convegno di Idraulica e Costruzioni Idrauliche, Brescia*.

Radice A., Rosatti G., Ballio F., Franzetti S., Mauri M., Spagnolatti M., Garegnani G. (2013), "Management of flood hazard via hydro-morphological river modelling. The case of the Mallero in Italian Alps", *Journal of Flood Risk Management*, Vol. 6, n. 3, 197-209, doi: 10.1111/j.1753-318X.2012.01170.x.

Rosatti G., Fraccarollo L. (2006), "A well-balanced approach for flows over mobile-bed with high sediment-transport", Journal of Computational Physics, 220(1), 312–338, https://doi.org/10.1016/j.jcp.2006.05.012.

Rosatti G., Fraccarollo L., Armanini A. (2004), "Behaviour of small perturbations in 1d mobile-bed models", River Flow, 67–73 (2004).

Zanchi B., Radice A. (2021), "Celerity and height of aggradation fronts in gravel-bed laboratory channel", Journal of Hydraulic Engineering, Vol. 147, n. 10, 04021034, doi: 10.1061/(ASCE)HY.1943-7900.0001923.

---

## Referee Report (RR1)

**2nd Review: Investigating the celerity of propagation for small perturbations and dispersive sediment aggradation under a supercritical flow**

By: Hasan Eslami, Erfan Poursoleymanzadeh, Mojtaba Hiteh, Keivan Tavakoli, Melika Yavari Nia, Ehsan Zadehali, Reihaneh Zarrabi, and Alessio Radice

**General comments**

Thank you for addressing my previous comments. I can clearly see that the article has improved significantly in its presentation, explanations, analysis, and discussion compared to the previous version. I appreciate the detailed manner in which my concerns were addressed. At this point, I only have a few minor comments that I believe are important to address before this paper is published.

Detailed comments:

1) I believe that associating celerity with eigenvalues is an excellent approach. However, for readers who are not entirely familiar with these concepts, I suggest including a schematic figure or cartoon that illustrates what the celerity of the bed surface represents. This should be Figure 1, as it would help all readers understand the phenomenon you are describing. I recommend showing a front moving downstream; even though this is more representative of subcritical conditions, it would provide a contextual framework for the entire paper.

2) Abstract – Line 25: "Scatter plots indicate…" As mentioned in the first review, plots are tools used to analyze data, but they don't "indicate" something on their own. Rather, it is your interpretation of the plots that indicates a process. Consider revising this wording to more accurately reflect the relationship between data visualization and analysis.

3) Line 69: "prior investigations…" This sentence appears to be missing the word "only" to properly convey your intended meaning.

4) Line 125: The text mentions that Equation 1 is valid for a unit-width rectangular channel. While this is correct, this limitation—specifically the rectangular channel assumption—should be placed in context in the introduction and acknowledged there as well. As the article is currently presented, there is no mention of this constraint earlier in the paper. Furthermore, since engineering applications are mentioned in the abstract, it's important to note that natural channels may differ significantly from rectangular channels, which could affect the applicability of the findings.

5) "water ripples, mentioned above, …" This reference is incorrect, as the discussion of water ripples appears in another section, not "above" this point in the text. Please modify

this reference to accurately direct the reader to the appropriate section where the simple cases are discussed.

6) Line 178: "This finding is somehow consistent..." This statement is vague and imprecise. The authors should explicitly state in what specific ways the findings are consistent with previous research or expectations, rather than using the ambiguous term "somehow." Clear articulation of the relationship between current and previous findings would strengthen this discussion.

7) Line 261: "... under the assumption of uniform flow..." I question the validity of this assumption in the context of supercritical flow with rapidly varying bed elevation. This potential limitation should be explicitly discussed in the paper, as non-uniform flow conditions may significantly affect the applicability of the presented analysis in such dynamic environments.

8) Figure 2: Changes in water depth along the channel clearly shown in this figure demonstrate that this is not uniform flow. Please discuss the potential errors associated with the uniform flow assumption and provide appropriate justification for its use despite this visual evidence to the contrary. An assessment of how these errors might impact your results and conclusions would strengthen the validity of your analysis.

9) "as equal to 1 s and 1.8 cm..." Please explain how these specific values were determined or selected, and clarify what they represent in the context of your analysis. Providing the rationale behind these parameter choices would enhance the reproducibility of your study and help readers better understand your methodological approach.

10) Line 367: "For the sake of ... of Fig 5 and 6" This sentence referencing Figure 4 is out of place and interrupts the flow of the paragraph. Please move it to where it logically belongs in the text to maintain coherence in your discussion.

11) Conclusion – Line 482: "(iv) The celerity..." Is this value ($10^{-2}$ times the water velocity) the "rule of thumb" mentioned in the abstract? The origin of this value should be clearly explained. If I missed the derivation in the paper, please ensure it is explicitly presented, as this represents the engineering application mentioned in the abstract and would be particularly valuable for practitioners. This practical insight should be well-substantiated and clearly communicated given its prominence in the abstract.

---

## Author Response (AR2)

Manuscript EGUSPHERE-2024-414: Investigating the celerity of propagation for small perturbations and dispersive sediment aggradation under a supercritical flow
Hasan Eslami et al.

**Response to Reviewers' and Editor's comments**

March 1st, 2025

Dear Editor, we submit the R2 version of our manuscript. All the Reviewers' issues have been considered, and either accepted or rebutted. This letter contains detailed responses to all the comments and suggestions received.

**Decision by Jens Turowski**

Associate editor decision: Publish subject to minor revisions (review by editor)

Dear authors,
thanks for the thorough revisions. Both original reviewers have looked at the paper again and are happy with the changes in content. However, both of them commented on readability, and sometimes confusing or difficult to follow writing (Reviewer #2 mentions this in his comment to the editor, rather than directly in the comment to the authors). Reviewer #1 makes some concrete suggestions for improvements.
I generally agree with the assessment and I return the paper to you for minor revisions, and ask you to go through the paper with a focus on language, clarity of argument, and structure. A lot of points I would make here are on style, and therefore somewhat a personal choice. However, premit me to make some general remarks on style (I leave it up to you in how far you want to take this up):
- Introduction: for me the introduction's point is to motivate the research and demonstrate its necessity. Generally, this moves from an undisputed, broad statement in the opening paragraph towards the specific research question, using the literature to support the arguments. The last paragraph of the introduction introduces the aim and objectives of the study, and the approach to tackle them. I would not refer to the study and its aims before.

> We have removed the mentions to the present study and its aims from the first paragraphs of the Introduction. Thus, in the revised manuscript, scope and aims of the present study are outlined only in the last paragraph.

- sentences like 'Figure X shows parameter A plotted vs. parameter B' are often inefficient and almost never necessary (a point also made by reviewer #1). Instead, state the message that the figure is meant to support and refer to it in parentheses.

> We have rephrased the presentation of the Figures in the cases where it streamlined the sentences.

- the discussion may benefit from some sub-structure. I generally aim for three separate parts / themes: a discussion / appraisal of the approach and methods, an interpretation of the results, and a placement into the body of previously existing knowledge.

> We have divided the Discussion into three sub-sections.

- the conclusion is often read before the bulk of the paper and it is often argued that it should be understandable on its own. It generally summarizes the findings in a paragraph and then provides the conclusions and wider implications.

> We have added a couple of sentences to the Conclusions section (at the beginning of the section), to make it stand-alone.

I hope this helps and I am looking forward to your revised paper.
All the best wishes, Jens Turowski

> Many thanks for handling the manuscript, the supportive decision and the useful suggestions.

**Comments by Angel Monsalve (Reviewer 1)**

Thank you for addressing my previous comments. I can clearly see that the article has improved significantly in its presentation, explanations, analysis, and discussion compared to the previous version. I appreciate the detailed manner in which my concerns were addressed. At this point, I only have a few minor comments that I believe are important to address before this paper is published.

Many thanks for recognizing our effort and for the additional suggestions.

1) I believe that associating celerity with eigenvalues is an excellent approach. However, for readers who are not entirely familiar with these concepts, I suggest including a schematic figure or cartoon that illustrates what the celerity of the bed surface represents. This should be Figure 1, as it would help all readers understand the phenomenon you are describing. I recommend showing a front moving downstream; even though this is more representative of subcritical conditions, it would provide a contextual framework for the entire paper.

We have added a Fig. 1 with sketches for translational and dispersive aggradation.

2) Abstract – Line 25: "Scatter plots indicate..." As mentioned in the first review, plots are tools used to analyze data, but they don't "indicate" something on their own. Rather, it is your interpretation of the plots that indicates a process. Consider revising this wording to more accurately reflect the relationship between data visualization and analysis.

We have changes "Scatter plots" into "Our results".

3) Line 69: "prior investigations..." This sentence appears to be missing the word "only" to properly convey your intended meaning.

Added. In the revised manuscript, the statement has been moved to the last paragraph of the Introduction.

4) Line 125: The text mentions that Equation 1 is valid for a unit-width rectangular channel. While this is correct, this limitation—specifically the rectangular channel assumption— should be placed in context in the introduction and acknowledged there as well. As the article is currently presented, there is no mention of this constraint earlier in the paper. Furthermore, since engineering applications are mentioned in the abstract, it's important to note that natural channels may differ significantly from rectangular channels, which could affect the applicability of the findings.

Formulating the system of the Saint Venant and Exner equations in this way is functional to writing it in vector form and then determining the eigenvalues. We have added this explanation to the paragraph between eq. (1) and eq. (2).

5) "water ripples, mentioned above, ..." This reference is incorrect, as the discussion of water ripples appears in another section, not "above" this point in the text. Please modify this reference to accurately direct the reader to the appropriate section where the simple cases are discussed.

We have changed "above" into "in the Introduction".

6) Line 178: "This finding is somehow consistent..." This statement is vague and imprecise. The authors should explicitly state in what specific ways the findings are consistent with previous research or expectations, rather than using the ambiguous term "somehow." Clear articulation of the relationship between current and previous findings would strengthen this discussion.

The paragraph just below eq. (12) has been rephrased to follow the Reviewer's suggestion.

7) Line 261: "... under the assumption of uniform flow..." I question the validity of this assumption in the context of supercritical flow with rapidly varying bed elevation. This potential limitation should be explicitly discussed in the paper, as non-uniform flow conditions may significantly affect the applicability of the presented analysis in such dynamic environments.

In the revised manuscript, we have clarified that the use of the uniform-flow equation is just functional to quantify reference parameters of the experiment, where depth and velocity

continuously vary in space and time as aggradation proceeds. Text was revised just above Table 1.

8) Figure 2: Changes in water depth along the channel clearly shown in this figure demonstrate that this is not uniform flow. Please discuss the potential errors associated with the uniform flow assumption and provide appropriate justification for its use despite this visual evidence to the contrary. An assessment of how these errors might impact your results and conclusions would strengthen the validity of your analysis.

The previous reply should have clarified that the assumption of uniform flow is used to provide a reference value for the initial flow condition. Indeed, if we look at the 5-s profiles in Fig. 2, the uniform flow is reasonably approximated.

9) "as equal to 1 s and 1.8 cm..." Please explain how these specific values were determined or selected, and clarify what they represent in the context of your analysis. Providing the rationale behind these parameter choices would enhance the reproducibility of your study and help readers better understand your methodological approach.

Explanation has been added below eq. (20).

10) Line 367: "For the sake of ... of Fig 5 and 6" This sentence referencing Figure 4 is out of place and interrupts the flow of the paragraph. Please move it to where it logically belongs in the text to maintain coherence in your discussion.

Since the smoothed version of the color gradient map of the Froude number is not presented in the manuscript, we believe that this is the appropriate place for this statement.

11) Conclusion – Line 482: "(iv) The celerity..." Is this value ($10^{-2}$ times the water velocity) the "rule of thumb" mentioned in the abstract? The origin of this value should be clearly explained. If I missed the derivation in the paper, please ensure it is explicitly presented, as this represents the engineering application mentioned in the abstract and would be particularly valuable for practitioners. This practical insight should be well-substantiated and clearly communicated given its prominence in the abstract.

At the end of the Discussion section we have added a reference to the Figs. 9 and 11 (new numbers after adding Fig. 1), to corroborate a statement that "$C/u$ was less than 0.05". Furthermore, we have mentioned this value explicitly in the abstract of the revised manuscript, and rephrased the last point of the Conclusions.

**Comments by Reviewer 2**

The authors did a good job in revising this manuscript. However, the language and presentation are not clear to convey the messages as well as deliver scientific insight to the audience. As a reader, I find it difficult to follow. For example, method and results are mixed up in section 4. Raw results should be moved to the result section.

Many thanks for recognizing our effort and for the additional suggestions.

While respecting the point of view of the Reviewer, we have finally resolved to leave the raw results in section 4. This is because the profiles of the bed elevation are useful to understand the application of eq. (20) that is crucial for the determination of the aggradation wave celerity. This is a recurring issue in papers, where a description of the methods without seeing any result remains, to our opinion, too abstract and difficult to follow. In order to clarify our intention, we have added (just below Table 1) a statement that raw results are presented in a method section to make following methods more easily understandable.

In the result section, several sentences start with "Figure presents/shows something" without giving much intuition about them.

We have rephrased the presentation of the Figures in the cases where it streamlined the sentences, as also suggested by the AE.

You can, for example, link the evolution of the celerity with changes in hydraulics during the experiments. You can also choose a specific location along the bed profile to show how the celerity of aggradation wave dynamically evolves with respect to the controlled hydraulics of the flume.

> The second comment would imply adding extra figures; a deeper phenomenological investigation of the celerity dynamics would surely be interesting, but beyond the scope of the paper and beyond the level of revision we were asked for. We have a further paper in our pipeline, that will be indeed focused on the celerity of the aggradation wave as a fundamental quantity for the process.

Several figures are in poor quality (e.g., figures 3, 4, 5 ,6, 7); the color map and color bar are difficult to visually discern.

> In the color gradient maps, we used multicolor scales instead of a single-band gradation that would have been harder to inspect. While we concur with the Reviewer that readability of color maps is generally an issue, all our maps have scatter-plot counterparts that are, instead, much more friendly for the human eye.

I think discussion is the weak section, as I do not see much interpretation of the celerity of small propagation as well as aggradation waves and implications for use cases. The first two paragraphs of the discussion are essentially results. Where should the reader find the discussion of the first type of correlation (Fr- lambda/u)?

> In order to account for this comment, some lines have been added to the first paragraph of section 6.2.

I could not quite follow the discussion until the last paragraph of the engineering point of view. Therefore, I highly recommend the author use simple and concise language, and a logical structure to display the results and discussion.

> We have divided the Discussion into three sub-sections to better guide the reader through it.

I have learned a usefully strategy that you generally can use the 4-step model to structure abstracts: 1) background and motivation, 2) research gap or question, 3) approach and method, 4) outcome and implications. Lines 20-23 appear to be redundant to me; what is the rationale for including supplementary information in the abstract? Line 25, perhaps, change "scatter plots" into "our results" which can be more compelling. Line 27, can you be more specific on mentioning the bulk value constrain? For instance, you might consider stating that $C/u < 0.05$ according to your results.

> The structure of our abstract follows the steps mentioned by the Reviewer, apart from declaring the manuscript scope in the very first line (which is an approach that others frequently recommend). Furthermore, in the revised manuscript we have followed the specific suggestions of the Reviewer.

There are several typos in the manuscript and misuse of language; kindly proofread and correct them again. For instance, it may be better to remove the word "for" on line 48; and on line 53, you might consider employing the term 'causes'; on line 432, I don't understand the point here.

> We have accepted the first two suggestions and tried to improve the argument mentioned in the third comment. Furthermore, text has been polished at several places along the manuscript.